# SpikSSD: Better Extraction and Fusion for Object Detection with Spiking Neuron Networks

## Abstract

As the third generation of neural networks, Spiking Neural Networks (SNNs) have gained widespread attention due to their low energy consumption and biological interpretability. Recently, SNNs have made considerable advancements in computer vision. However, efficiently conducting feature extraction and fusion under the spiking characteristics of SNNs for object detection remains a pressing challenge. To address this problem, we propose the SpikSSD, a novel Spiking Single Shot Multibox Detector. Specifically, we design a full-spiking backbone network, MDS-ResNet, which effectively adjusts the membrane synaptic input distribution at each layer, achieving better spiking feature extraction. Additionally, for spiking feature fusion, we introduce the Spiking Bi-direction Fusion Module (SBFM), which for the first time realizes bi-direction fusion of spiking features, enhancing the multi-scale detection capability of the model. Experimental results show that SpikSSD achieves 40.8% mAP on the GEN1 dataset, 76.3% and 52.4% mAP@0.5 on VOC 2007 and COCO 2017 datasets respectively with the lowest firing rate, outperforming existing SNN-based approaches at ultralow energy consumption. This work sets a new benchmark for future research in SNN-based object detection. Our code is publicly available in supplementary materials.

## 1 Introduction

Spiking Neural Networks (SNNs), unlike traditional Artificial Neural Networks (ANNs) that rely on continuous values, communicate between neurons through discrete spiking signals and use biologically interpretable spiking neurons for computation (Maass (1997)). Although the discreteness of spiking signals may lead to information loss, the temporal property of spikes can compensate for this limitation (Skatchkovsky et al. (2021)). Additionally, due to the spiking nature of SNNs, they can avoid the heavy computation load of multiplication operations, relying instead on simpler addition operations (Hu et al. (2024)). These characteristics give significant advantages to SNNs in terms of energy efficiency and processing speed, positioning them as a promising alternative to ANNs.

Object detection is a crucial task in computer vision, with wide-ranging applications from autonomous driving (Balasubramaniam & Pasricha (2022)) to intelligent surveillance (Zhang & Kim (2019)) and medical imaging (Litjens et al. (2017)). Moreover, object detection is not only a classification task but also involves regression, requiring the simultaneous identification of object categories and the precise prediction of their locations and sizes.

Recently, SNNs have made significant progress in classification tasks (Zhou et al. (2024); Yao et al. (2023)). However, their performance in more complex tasks that involve both classification and regression, such as object detection, still falls short of expectations. In this situation, many studies on SNN-based object detection have focused on improvements to the backbone (Su et al. (2023)) and introducing the spiking feature fusion network (Fan et al. (2024)). While these approaches have achieved some success, they fail to consider the network as a whole and instead focus only on optimizing specific parts. To enhance the performance of SNNs in detection tasks, it is essential to focus on both efficient extraction and fusion of spiking features, as a strong backbone requires effective fusion to utilize its features fully, and vice versa. Therefore, this paper proposes SpikSSD (Spiking Single Shot Multibox Detector) to address these challenges.

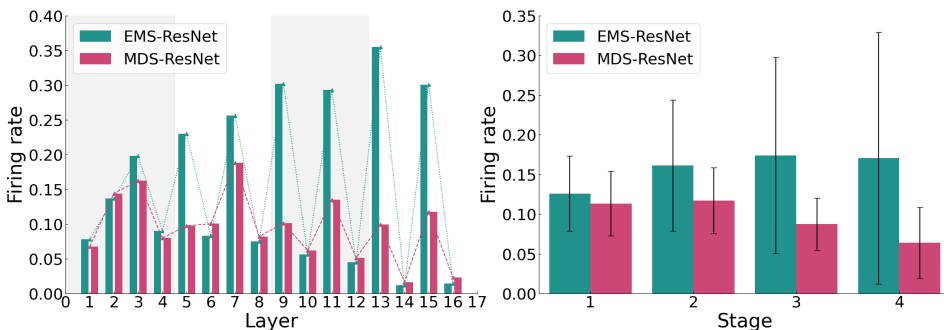

Figure 1: **Firing patterns of MDS-ResNet and EMS-ResNet.** Left: Bar chart and line graph showing firing rates for each layer. Right: Bar chart with standard deviation displaying statistical features of both networks per stage. For a fair comparison, both networks use the ResNet18 configuration on the GEN1 dataset, and the firing rates shown are exclusively from the residual paths of each network.

For feature extraction, we observe that the most significant difference between SNNs and ANNs lies in their feature representation: in SNNs, features are coded through neuron firing patterns. In EMS-ResNet (Su et al. (2023)), the output of the residual path continuously accumulates to the shortcut path. Assuming these outputs are approximately independent, the variance of their sum equals the sum of their variances, leading to a gradual increase in membrane synaptic input variance for subsequent neurons. This results in a higher probability of extreme inputs. Specifically, when the synaptic input is small, the accumulated membrane potential fails to reach the firing threshold. When it is excessively large, neurons exhibit abnormally high firing rates. In both cases, the firing pattern becomes either overly sparse or excessive, failing to reflect the input features accurately. This unstable firing pattern makes it difficult for the network to distinguish and extract meaningful features, which in turn affects the network's feature extraction capability.

To address this issue, we propose the Membrane-based Deformed Shortcut (MDS), a novel mechanism optimizing identity mapping to stabilize membrane synaptic input distribution across layers. Based on MDS, we propose a novel Spiking Residual Network, MDS-ResNet, which exhibits stronger feature extraction capabilities. The firing patterns of MDS-ResNet and EMS-ResNet are shown in Fig. 1, where it can be seen that our MDS effectively stabilizes the firing pattern.

In SNNs, features exist in both spatial and temporal domains. Spiking feature fusion enhances feature representation in both dimensions, improving multi-scale detection capabilities. This dual-domain nature makes feature fusion more crucial in SNNs compared to ANNs. Currently, the only fusion method designed for SNNs is the Spiking Fusion Module (SFM) (Fan et al. (2024)). This fusion method adopts a one-way, down-up approach, where large-scale features are upsampled to merge with small-scale features. Without up-down fusion, it only enhances small-scale feature representation, overlooking the potential improvement of large-scale features. Furthermore, SFM incorporates SEW-Block internally, which compromises the spiking nature of the network. To address these issues, we propose the Spiking Bi-direction Fusion Module (SBFM). Specifically, this module utilizes Spiking Up/Down Block and membrane addition-based fusion method to process feature maps, performing fusion first bottom-up and then top-down. We employ the proposed MDS Fusion Block (MDSF-Block) to extract fused features during this process. These SNN-friendly components ensure SBFM adheres to spiking characteristics while enhancing the model detection capabilities.

The main contributions of this work can be summarized as follows:

- We propose MDS-ResNet, which introduces the MDS to stabilize membrane synaptic input distribution at each layer, thereby enhancing feature extraction capabilities. Additionally, we theoretically demonstrate that MDS-ResNet can avoid gradient vanishing or explosion.

- We propose the SBFM, which achieves bi-directional feature fusion for the first time in SNNs, thereby enhancing the model's capability to detect multi-scale objects.

- Based on MDS-ResNet, SBFM, and the SSD Detection Head, we propose SpikSSD, which achieves state-of-the-art mAP on event-based GEN1, frame-based VOC 2007 and COCO 2017 datasets compared to other SNN-based models, while requiring the lowest firing rate.

## 2 RELATED WORK

### 2.1 SPIKING NEURAL NETWORKS

SNNs are designed to mimic the behavior of biological neurons more accurately than ANNs. Researchers introduce various spiking neural models to achieve this, including the Hodgkin-Huxley (Hodgkin & Huxley (1952)), Izhikevich (Izhikevich (2003)), Leaky Integrate-and-Fire (LIF) (Wu et al. (2019)), and Parametric Leaky Integrate-and-Fire (PLIF) (Fang et al. (2021b)) models.

SNNs utilize two training approaches: ANNs-to-SNNs conversion and direct training. The ANNs-to-SNNs conversion approach approximates ReLU activation using average firing rate, allowing trained ANNs to be transformed into SNNs (Cao et al. (2015)). Although this approach has led to powerful SNNs (Kim et al. (2020b)), it requires thousands of time steps and prevents SNNs from exploiting their low-latency advantage. Moreover, this method is suited only for static datasets and the performance of converted SNNs depends on the original ANNs. Conversely, the direct training method leverages surrogate gradients to optimize SNNs (Neftci et al. (2019)), facilitating their training on a variety of datasets and achieving strong performance within a limited number of time steps. This technique enables the widespread application of SNNs in vision tasks (Cordone et al. (2022); Zhu et al. (2022)). Hence, we apply the direct training strategy to our model in this study.

### 2.2 SPIKING RESIDUAL NETWORKS

In the Spiking Residual Networks (SRNs) proposed based on the ANN-to-SNN conversion training method, the primary focus is ensuring that the SNNs maintain lossless accuracy during the conversion from ANNs (Hu et al. (2021)). However, networks designed based on this method also face inherent drawbacks associated with this training approach.

In directly trained SRNs, Zheng et al. (2021) successfully constructs a 50-layer deep network using threshold-dependent Batch Normalization (tdBN), but overlooks spiking characteristics by simply replacing ReLU with LIF neurons. SEW-ResNet (Fang et al. (2021a)) and MS-ResNet (Hu et al. (2024)) address gradient issues and train networks exceeding 100 layers. However, the spike addition for residual learning in SEW-ResNet and the neglect of non-spiking convolutions for shortcuts by MS-ResNet prevent full-spiking networks. EMS-ResNet Su et al. (2023) addresses the shortcomings of MS-ResNet but fails to improve performance due to inadequate consideration of SNN characteristics. This paper further investigates spiking characteristics to enhance feature extraction while maintaining a full-spiking SRN.

### 2.3 OBJECT DETECTION WITH SPIKING NEURON NETWORKS

Unlike ANNs, SNNs have event-driven characteristics suitable for both frame and event cameras. Event cameras have garnered attention due to their advantages in temporal resolution, dynamic range, power efficiency, and pixel bandwidth (Gallego et al. (2020)).

Early attempts to apply SNNs to object detection using ANN-to-SNN conversion (Kim et al. (2020b;a)) face challenges of high latency, poor performance, and incompatibility with event cameras. VC-DenseNet (Cordone et al. (2022)) and EMS-YOLO (Su et al. (2023)) pioneer direct training for SNN-based object detection, achieving promising results on both frame and event-based datasets. However, they both overlook feature fusion. SFOD (Fan et al. (2024)) introduces an efficient Spiking Fusion Module, but emphasizes less on backbone feature extraction. Hence, our research enhances SNN-based object detection by improving both feature extraction and fusion.

## 3 METHOD

### 3.1 OVERVIEW

The architecture of our proposed SpikSSD is shown in Fig. 2. For event data, we use the voxel cube (Cordone et al. (2022)) for coding. Compared to the voxel grid (Bardow et al. (2016)), this method can store more temporal information in the channel dimension, thereby improving the efficiency

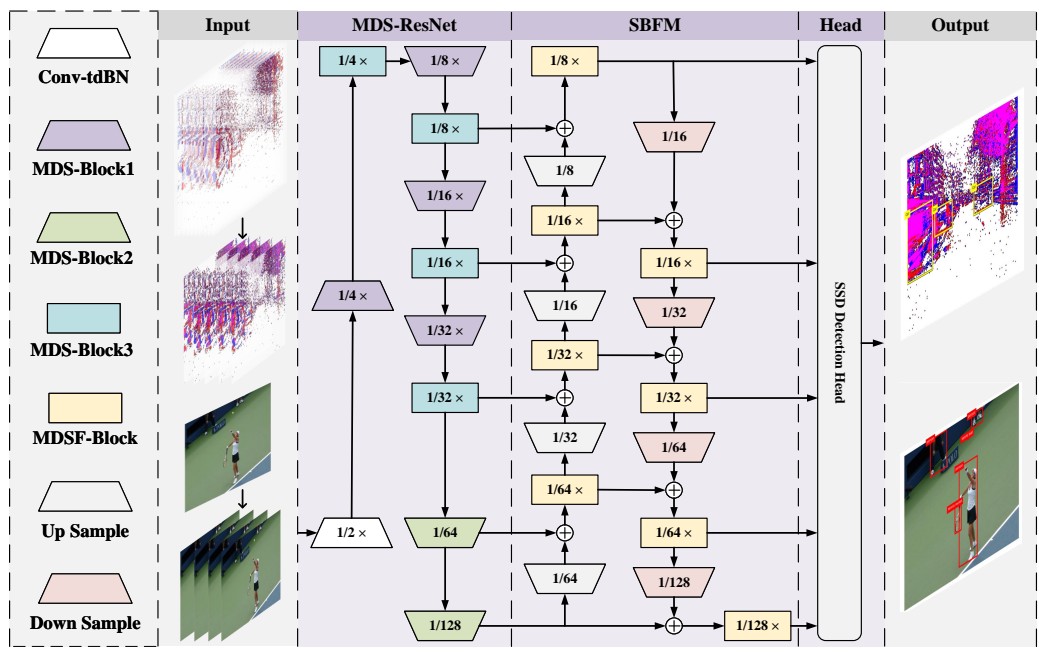

Figure 2: **The architecture of SpikSSD.** SpikSSD comprises MDS-ResNet, SBFM, and the SSD Detection Head. The backbone is illustrated using the ResNet18 configuration He et al. (2016). The total downsampling factors for each block are annotated in the figure. The model receives two types of inputs: event and static data. The input coding and output are represented in the figure.

in processing sparse and asynchronous event data. For static data, to leverage the spatiotemporal characteristics of SNNs, we employ the direct coding method (Kim et al. (2022)), replicating the input $T$ times to form a time train that supports the temporal processing requirements of the model.

Then, we feed the coded data into MDS-ResNet for feature extraction. To capture more multi-scale information, two additional MDS-Block2 are integrated after the backbone network. When feature extraction is completed, features with resolutions ranging from 1/8 to 1/128 are processed through the Spiking Bi-direction Fusion Module (SBFM) for feature fusion. This module employs a down-up followed by an up-down bi-directional fusion method. The resulting fused feature maps are then fed into the SSD Detection Head (Liu et al. (2016)) for object detection. Finally, softNMS (Bodla et al. (2017)) is used for post-processing to optimize detection results.

Notably, the SSD Detection Head (SDH) consists of only a single convolution layer. During the training phase, to aid in the model convergence, we first decode the data using spiking rate decoding before inputting them into the SDH (Fan et al. (2024)). In the inference phase, we divide the SDH parameters by the number of time steps T and directly input the spike train into the SDH, finally performing summation over the time dimension. This strategy ensures consistency between the inference and training results while preserving the spiking characteristics of the model.

### 3.2 MDS-RESNET

Currently, the main deep Spiking Residual Networks include MS-ResNet and EMS-ResNet. For MS-ResNet, its residual learning can be expressed as follows.

$$\psi = \text{tdBN} \circ \text{Conv} \circ \text{SN} \tag{1}$$

$$Y_I^L = \psi\left(\psi(X^L)\right) + X^L \tag{2}$$

$$Y_D^L = \psi\left(\psi(X^L)\right) + \text{tdBN}(\text{Conv}(X^L)) \tag{3}$$

Here, $Y$ represents output and $X$ represents input. The superscript $L$ denotes the $L$-th residual block, and the subscripts $D$ and $I$ indicate whether the residual block performs downsampling or

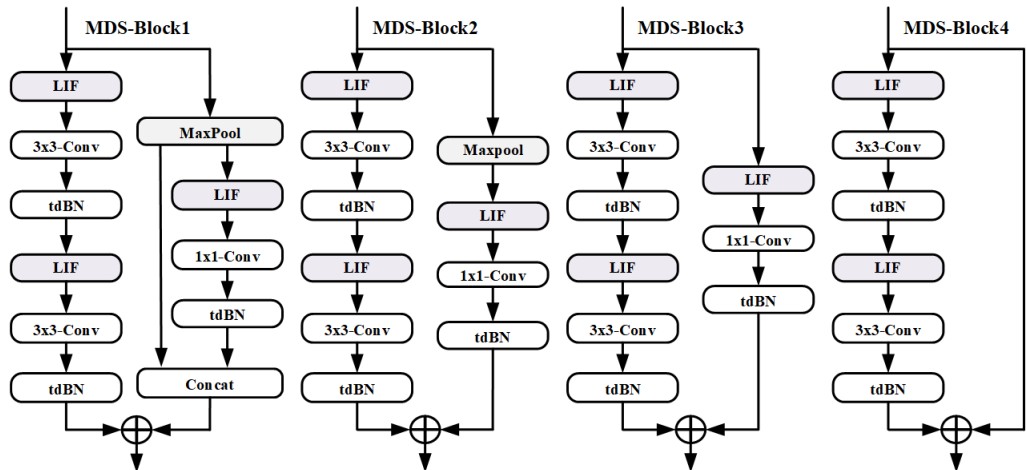

Figure 3: **Architecture of the four blocks in MDS-ResNet.**

not, respectively. SN, tdBN, and Conv denote spiking neurons, threshold-dependent Batch Normalization, and convolution, respectively. From the above equations, it can be seen that in shortcuts involving downsampling, convolution is performed on membrane synaptic input without converting them to spikes. This disrupts the spiking characteristics of SNNs and introduces a substantial amount of non-spiking operations.

To address this issue, EMS-ResNet improves $Y_D^L$, with its residual learning formula as follows. Here, $\mathrm{MaxPool}$ represents Maxpool, $Y_{D1}^L$ represents downsampling with an increased number of channels, and $Y_{D2}^L$ represents downsampling with the same or decreased number of channels. Through this improvement, EMS-ResNet effectively preserves the spiking characteristics of SNNs.

$$Y_{D1}^L = \psi\left(\psi(X^L)\right) + \mathrm{Concat}[\mathrm{MaxPool}(X^L), \psi(\mathrm{MaxPool}(X^L))] \tag{4}$$

$$Y_{D2}^L = \psi\left(\psi(X^L)\right) + \psi(\mathrm{MaxPool}(X^L)) \tag{5}$$

However, we believe that the $Y_I^L$ in MS-ResNet and EMS-ResNet does not fully consider the characteristics of SNNs that utilize firing patterns for feature representation. For ease of analysis, let $Y_R^L$ denote the residual path output and $Y_S^L$ denote the shortcut path output. We assume the distributions of these two variables to be $Y_R^L \sim \mathcal{N}(0, \sigma_R^{L^2})$ and $Y_S^L \sim \mathcal{N}(0, \sigma_S^{L^2})$, where $\sigma_R^{L^2}$ and $\sigma_S^{L^2}$ represent their respective variances. Assuming the two variables are approximately independent, $Y_I^L \sim \mathcal{N}(0, \sigma_R^{L^2} + \sigma_S^{L^2})$. Since there is no computation on the shortcut path, we can derive that $Y_I^L \sim \mathcal{N}(0, \sigma_R^{L^2} + \sigma_R^{L-1^2} + \sigma_S^{L-1^2})$. Therefore, the variance of $Y_I^L$ becomes increasingly large as the network deepens. This results in unstable membrane synaptic input distribution in subsequent neurons, leading to unstable cumulative membrane potential distributions. Consequently, the probability of overly sparse or excessive firing patterns increases, which impedes accurate feature representation and ultimately weakens the feature extraction capability of the model.

While adding tdBN at the shortcut could address this problem, it introduces non-spiking operations, compromising the inherent characteristics of SNNs. To preserve these characteristics while solving the issue, we propose the Membrane-based Deformed Shortcut (MDS), which incorporates LIF-Conv1x1-tdBN. This approach employs a LIF layer to convert input into spikes, followed by a 1x1 convolution layer. The convolution not only enhances the expressiveness of the network but also enables batchnorm-scale-fusion with tdBN during inference (Zheng et al. (2021)). Finally, the tdBN effectively adjusts the distribution of the shortcut output. Building upon the MDS, we propose MDS-ResNet, a new variant of SRN that incorporates MDS into the network architecture.

The proposed MDS-ResNet comprises four blocks, with their structures illustrated in Fig. 3. MDS-Block1 and 2 are downsampling blocks, adopting the idea of EMS-ResNet. MDS-Block3 and 4 are designed without downsampling. MDS-Block3 incorporates MDS in its shortcut, effectively adjusting the output variance. MDS-Block4, identical to MS-Block (Hu et al. (2024)), facilitates gradient propagation. This combination of the two block varieties prevents degradation of gradient

flow in deeper networks, which could occur if only MDS-Block3 is used throughout the model. Notably, while some feature maps are directly added to the residual path in the shortcuts of MDS-Block1 and 4, MDS-Block2 and 3 allow for quick adjustment of membrane synaptic input variance, ensuring effective feature extraction capabilities across the network.

### 3.3 ANALYSIS OF GRADIENT VANISHING/EXPLOSION FOR MDS-RESNET

To demonstrate that the MDS does not affect the training and convergence, we use the Block Dynamical Isometry (Chen et al. (2020)) to verify that MDS-ResNet can effectively overcome the gradient vanishing or explosion problem.

Consider a serial network as shown in Eq. (6), where $\mathbf{f_j}$ represents the j-th layer of the network. The jacobian matrix from input to output of the j-th layer can be denoted as $\mathbf{J_j} = \frac{\partial \mathbf{f_j}}{\partial \mathbf{f_{j-1}}}$. Furthermore, let $\phi(\mathbf{J})$ be defined as the expectation of $\text{tr}(\mathbf{J})$, while $\varphi(\mathbf{J}) = \phi(\mathbf{J}^2) - \phi^2(\mathbf{J})$

$$\mathbf{f}(\mathbf{x}) = \mathbf{f_L} \circ \mathbf{f_{L-1}} \circ \cdots \circ \mathbf{f_1}(\mathbf{x}) \tag{6}$$

**Lemma 1.** (Chen et al. (2020)) *Consider a neural network that can be represented as a series of blocks as Eq. (6) and the j-th block's jacobian matrix is denoted as $\mathbf{J_j}$. If $\forall j$, $\phi(\mathbf{J_j}\mathbf{J_j}^T) \approx 1$ and $\varphi(\mathbf{J_j}\mathbf{J_j}^T) \approx 0$, the network achieves Block Dynamical Isometry and can avoid gradient vanishing or explosion.*

**Definition 1. (General Linear Transform)** (Chen et al. (2020)) *Let $\mathbf{f}(\mathbf{x})$ be a transform whose Jacobian matrix is $\mathbf{J}$. $\mathbf{f}$ is called general linear transform when it satisfies:*

$$E\left[\frac{\|\mathbf{f}(\mathbf{x})\|_2^2}{len(\mathbf{f}(\mathbf{x}))}\right] = \phi\left(\mathbf{J}\mathbf{J}^T\right) E\left[\frac{\|\mathbf{x}\|_2^2}{len(\mathbf{x})}\right]. \tag{7}$$

**Lemma 2. (Multiplication)** (Chen et al. (2020)) *Given $\mathbf{J} := \prod_{j=L}^{1} \mathbf{J_j}$, where $\{\mathbf{J_j} \in \mathbb{R}^{m_j \times m_{j-1}}\}$ is a series of independent random matrices. If $(\prod_{j=L}^{1} \mathbf{J_j})(\prod_{j=L}^{1} \mathbf{J_j})^T$ is at least the $1^{st}$ moment unitarily invariant, we have*

$$\phi\left((\prod_{j=L}^{1} \mathbf{J_j})(\prod_{j=L}^{1} \mathbf{J_j})^T\right) = \prod_{j=L}^{1} \phi(\mathbf{J_j}\mathbf{J_j}^T). \tag{8}$$

Since the information transmitted in the network can be considered as random variables (Poole et al. (2016)), $E\left[\frac{\|\mathbf{x}\|_2^2}{len(\mathbf{x})}\right]$ is regarded as the second-order moment of the input, denoted by $\alpha_2$. Building on the derivations in (Chen et al. (2020); Su et al. (2023); Zheng et al. (2021)), it is established that LIF, Conv, tdBN, Concatenation, and Maxpool each qualify as general linear transform. Therefore, we can independently analyze each MDS-Block based on Definition 1 and Lemma 2. By multiplying these analysis results, we derive the overall behavior of the MDS-ResNet.

**Proposition 1.** *For MDS-Block1, MDS-Block2, and MDS-Block3, when the block output follows $\mathbf{x} \sim \mathcal{N}(0, 1)$, each satisfies: $\phi(\mathbf{J_j}\mathbf{J_j^T}) \approx \frac{1}{\alpha_2^{j-1}}$.*

**Proposition 2.** *For MDS-ResNet, when the encoding layer output follows $\mathbf{x} \sim \mathcal{N}(0, 2^{N_{MDS\text{-}Block4}})$ and each block output follows $\mathbf{x} \sim \mathcal{N}(0, 1)$, the network satisfies: $\phi(\mathbf{J}\mathbf{J}^T) \approx 1$.*

**Proof.** The details can be found in the supplementary materials.

As elucidated in (Su et al. (2023); Hu et al. (2024)), in the majority of cases, $\phi(\mathbf{J}\mathbf{J}^T) \approx 1$ is sufficient to guarantee that the network avoids gradient vanishing or explosion. The condition $\varphi(\mathbf{J_j}\mathbf{J_j}^T) \approx 0$ is imposed to preclude the occurrence of accidental situations. Therefore, Proposition 1 and Proposition 2 provide theoretical support for the ability of MDS-ResNet to address the problem of gradient vanishing or explosion.

### 3.4 SPIKING BI-DIRECTION FUSION MODULE

Research in feature fusion for SNNs is limited to one-way down-up fusion (Fan et al. (2024)), where large-scale features are only upsampled to enhance small-scale features. This approach restricts up-down information transmission and leads to inadequate feature fusion across scales. Furthermore,

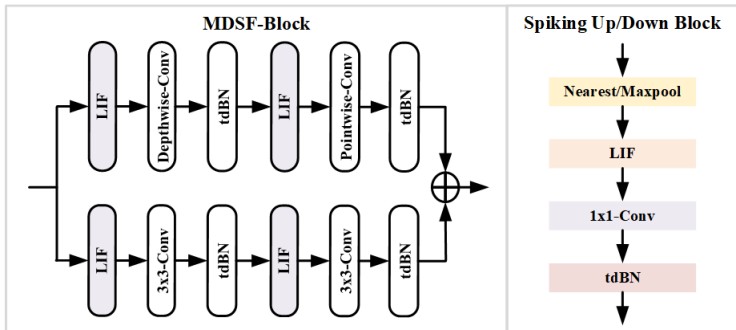

Figure 4: **The architecture of MDSF-Block and Spiking Up/Down Block.**

this method fails to preserve inherent spiking characteristics. To address these issues, we propose the Spiking Bi-direction Fusion Module (SBFM), which enhances feature fusion through bi-directional information flow while maintaining the network spiking properties.

The structure of the SBFM is shown in Fig. 2. In this module, feature maps ranging from 1/8 to 1/128 are fused. During the fusion process, we first fuse feature maps of lower resolution with those of higher resolution through upsampling (down-up). Subsequently, we employ downsampling further to integrate these fused feature maps (up-down). Notably, we perform only a single fusion operation for the 1/8 and 1/128 scale feature maps to avoid introducing feature maps at 1/4 and 1/256 scales, thereby enhancing computation efficiency. Next, we will discuss the methods for spiking feature fusion, upsampling, and downsampling.

The current spiking feature fusion method uses concatenation to preserve spiking characteristics (Fan et al. (2024)), but this increases model complexity and makes the effective alignment of features difficult. To address these issues, we propose a membrane addition-based fusion method that maintains spiking characteristics by adding membrane synaptic inputs. This method treats feature maps from different resolutions as supplements with smaller variances, adding them to the target-resolution feature map to mimic residual learning. The MDSF-Block (Fig. 4) enhances fusion through post-addition feature extraction. This block replaces MDS in MDS-Block3 with spiking depthwise separable convolution, enabling fusion across different dimensions while maintaining spike-based computation.

To achieve upsampling/downsampling, we propose the Spiking Up/Down Block, whose structure is shown in Fig. 4. In this block, we use Nearest Neighbor Interpolation or Maxpool as the key element, which has been proven to maintain the inherent spiking characteristics of SNNs during the upsampling/downsampling process (Rançon et al. (2022); Fang et al. (2021b)). Furthermore, to ensure that the feature maps after sampling are consistent in channel numbers with the feature maps to be fused, and to assign them a smaller variance, we incorporate a LIF-Conv1x1-tdBN structure following the Nearest Neighbor Interpolation or Maxpool operation.

## 4 EXPERIMENT

In this section, we conduct extensive ablation studies and analysis of the model on the GEN1 (De Tournemire et al. (2020)) dataset. We also compare the results with state-of-the-art methods on GEN1, PASCAL VOC (Everingham et al. (2010)), and COCO 2017 (Lin et al. (2014)) dataset.

### 4.1 EXPERIMENTAL SETUP

The membrane time constant $\tau$ for LIF/PLIF neurons is initialized to 0.25, with the threshold set to 1. The models are trained on 4 NVIDIA A6000 GPUs, using the AdamW optimizer with a learning rate of 1e-3 in conjunction with a cosine learning rate scheduler. On the Gen1 dataset, we employ the zoom-in and zoom-out augmentation strategies from (Gehrig & Scaramuzza (2023)). For PASCAL VOC, we train on the VOC 2007+2012 trainval and evaluate on the VOC 2007 test set. For both PASCAL VOC and COCO 2017 datasets, we use Mosaic and Mixup augmentation.

| Backbone | Fusion | mAP | mAP @0.5 | Param (M) | Firing Rate(%) | Energy (mJ) | Window (ms) | Input Scale |
|---|---|---|---|---|---|---|---|---|
| DenseNet121-24 | ✗ | 29.2 | 52.5 | 7.3 | 20.19 | 2.17 | 100 | 1.0 |
| SEW-ResNet18 | ✗ | 32.2 | 58.2 | 9.8 | 10.46 | 1.92 | 100 | 1.0 |
| EMS-ResNet18 | ✗ | 32.9 | 59.2 | 9.7 | 17.51 | 1.29 | 100 | 1.0 |
| MS-ResNet18 | ✗ | 34.1 | 60.4 | 9.8 | 12.80 | 2.04 | 100 | 1.0 |
| MDS-ResNet10 | ✗ | 29.3 | 54.6 | 6.6 | 18.48 | 0.80 | 100 | 1.0 |
| **MDS-ResNet18** | ✗ | **34.7** | **60.9** | **10.1** | **12.41** | **0.97** | **100** | **1.0** |
| **MDS-ResNet34** | ✗ | **37.4** | **64.1** | **15.2** | **12.41** | **1.73** | **100** | **1.0** |
| MDS-ResNet18 | ✓ | 35.7 | 62.3 | 13.9 | 10.58 | 1.47 | 50 | 1.0 |
| **MDS-ResNet18** | ✓ | **37.6** | **64.3** | **13.9** | **10.57** | **1.47** | **100** | **1.0** |
| **MDS-ResNet34** | ✓ | **39.2** | **66.5** | **19.0** | **11.56** | **2.32** | **100** | **1.0** |
| MDS-ResNet18 | ✓ | 38.0 | 64.9 | 13.9 | 10.40 | 1.44 | 200 | 1.0 |
| **MDS-ResNet18[†]** | ✓ | **38.2** | **65.3** | **13.9** | **11.22** | **1.56** | **200** | **1.0** |
| MDS-ResNet34[†] | ✓ | 39.7 | 67.2 | 19.0 | 12.20 | 2.45 | 200 | 1.0 |
| MDS-ResNet34[†] | ✓ | 39.5 | 66.9 | 19.0 | 11.67 | 2.90 | 200 | 1.1 |
| MDS-ResNet34[†] | ✓ | 40.7 | 68.0 | 19.0 | 11.93 | 3.42 | 200 | 1.2 |
| **MDS-ResNet34[†]** | ✓ | **40.8** | **68.8** | **19.0** | **11.59** | **3.97** | **200** | **1.3** |
| MDS-ResNet34[†] | ✓ | 40.5 | 68.1 | 19.0 | 10.11 | 4.01 | 200 | 1.4 |

Table 1: **Results of the ablation study on the GEN1 dataset.** A [†] indicates that the model adopts 100ms/200ms training/inference approach.

For object detection tasks, the main evaluation metrics are mAP (mAP@0.5:0.95) and mAP@0.5. The firing rate, which measures neuronal activity, is another critical metric for evaluating SNNs. It is calculated as the average ratio of neuron spikes to the total number of neurons across all time steps. Additionally, we report the energy consumption of the models. More implementation details and energy consumption calculation methods are provided in the supplementary materials.

## 4.2 ABLATION STUDIES

In this section, we first demonstrate the effectiveness of MDS-ResNet through a comparison with different spiking backbone networks. Next, we investigate the impact of model depth, the SBFM, the event time window, and the input scale on model performance.

### 4.2.1 DIFFERENT BACKBONE NETWORK

To better demonstrate the feature extraction capabilities of MDS-ResNet, we compare it with the most advanced SNN backbones currently available. For a fair comparison with DenseNet121-24 (Cordone et al. (2022)), the other networks are configured with ResNet18 (He et al. (2016)) to ensure similar parameter sizes. The comparison results are shown in rows 1 to 4 and 6 of Tab. 1. These results show that the MDS-ResNet18 outperforms other models in mAP and has a lower firing rate and energy consumption. This indicates that our proposed MDS effectively ensures the stability of membrane synaptic input, achieving optimal performance with the most efficient firing pattern.

### 4.2.2 IMPACT OF MODEL DEPTH

In the previous section, we theoretically demonstrate that the MDS-ResNet could overcome the issues of gradient vanishing or exploding, enabling the realization of large-scale networks. In this section, we further validate this through experiments. As shown in rows 5 to 7 of Tab. 1, the feature extraction capability of MDS-ResNet improves as the network depth increases.

### 4.2.3 THE EFFECTIVENESS OF SBFM

As shown in rows 9 and 10 of Tab. 1, the performance of MDS-ResNet18 and MDS-ResNet34 significantly improves after incorporating the SBFM. Specifically, not only does the mAP increase by about 2 points, but the firing rate also decreases, and energy consumption remains nearly unchanged.

| Model | Spike | Param (M) | mAP | Window (ms) | Firing Rate(%) | Energy (mJ) |
|---|---|---|---|---|---|---|
| Asynet (Messikommer et al. (2020)) | ✗ | 11.4 | 14.5 | 250 | - | >4.83 |
| MatrixLSTM (Cannici et al. (2020)) | ✗ | 61.5 | 31.0 | - | - | - |
| RED (Perot et al. (2020)) | ✗ | 24.1 | 40.0 | 50 | - | >24.08 |
| ASTMNet (Li et al. (2022a)) | ✗ | >100 | 46.7 | - | - | - |
| **RVT** (Gehrig & Scaramuzza (2023)) | ✗ | **18.5** | **47.2** | **50** | **-** | **-** |
| VC-DenseNet (Cordone et al. (2022)) | ✓ | 8.2 | 18.9 | 100 | 29.44 | 3.89 |
| KD-SNN (Bodden et al. (2024)) | ✓ | 13.0 | 22.9 | 100 | 17.40 | 1.00 |
| EMS-YOLO (Su et al. (2023)) | ✓ | 14.4 | 31.0 | 100 | 17.80 | - |
| SFOD (Fan et al. (2024)) | ✓ | 11.9 | 32.1 | 100 | 24.40 | 7.26 |
| **EAS-SNN** (Wang et al. (2024)) | ✓ | **25.3** | **37.5** | **200** | **-** | **28.10** |
| **SpikSSD-S** | ✓ | **13.9** | **39.0** | **200** | **10.90** | **2.58** |
| **SpikSSD-L** | ✓ | **19.0** | **40.8** | **200** | **11.59** | **3.97** |

Table 2: **Comparison with state-of-the-art models on the GEN1 dataset.**

This indicates that the SBFM enhances the model's multi-scale detection capability, achieving a good balance between effectiveness and energy efficiency.

### 4.2.4 INFLUENCE OF TIME WINDOW

For event data, the time window of samples varies, providing different bases for prediction. The larger the time window, the stronger the model becomes at detecting slowly moving objects. Here, we conduct thorough experiments by adjusting the size of the time window. As shown in rows 8 to 12 of Tab. 1, the model's mAP improves as the window size increases, while energy consumption remains constant. However, we argue that an unlimited expansion of the time window would impose an excessive burden on hardware memory, thus we do not experiment with larger time windows. Furthermore, as indicated in row 12, we discover that training the model with a 100ms window and inferring with a 200ms window yields excellent results. We attribute the improved performance to training with a 100ms window, which provides more sparse samples, thus enhancing model performance when inferring with a 200ms window. This approach reduces hardware burden and training time. Therefore, our final model adopts this 100ms/200ms training/inference approach.

### 4.2.5 INFLUENCE OF INPUT SCALE

We investigate the effect of scaling event data using Nearest Neighbor Interpolation, which has never been explored before. As shown in rows 13 to 17 of Tab. 1, model performance improves as the scale increases up to 1.3, but declines beyond this point. This decline is likely due to events becoming more spatially sparse at larger scales, creating challenges for the model inference.

### 4.3 BENCHMARK COMPARISONS

Based on ablation experiments, we propose a small model (SpikSSD-S) using MDS-ResNet18 as the backbone and a large model (SpikSSD-L) using MDS-ResNet34 as the backbone. We compare our models with other state-of-the-art models on the GEN1, VOC 2007, and COCO 2017 datasets.

On the GEN1 dataset, as shown in Tab. 2, our lightweight SpikSSD-S achieves a significantly higher mAP while having comparable parameters to other models. Notably, SpikSSD-S demonstrates superior performance compared to the full-spiking model of EAS-SNN, achieving a higher mAP while utilizing only half the parameters and consuming merely 1/10 of the energy. Our SpikSSD-L further improves performance, becoming the first full-spiking SNN model to exceed mAP of 40.0, while its energy consumption is less or comparable to other SNN-based models. Compared to ANN-based models, our model further narrows the gap with the state-of-the-art, offering significant advantages in energy consumption. The inference results of SpikSSD-L in comparison with SpikSSD-S and Ground Truth are presented in Fig. 5. On the VOC 2007 and COCO 2017 datasets, as demonstrated in Tab. 3, SpikSSD achieves state-of-the-art and second-best mAP@0.5 respectively, while consuming less energy compared to existing SNN-based methods. Notably, SpikSSD ranks first among directly-trained SNN-based methods on COCO 2017. Moreover, when compared to the ANN-based

| Method | Model | VOC 2007 | | | COCO 2017 | | |
|---|---|---|---|---|---|---|---|
| | | Param (M) | mAP @0.5 | Energy (mJ) | Param (M) | mAP @0.5 | Energy (mJ) |
| ANNs | YOLOv5s (Qu et al. (2024)) | **7.2** | **75.3** | - | 7.3 | 54.8 | 38.41 † |
| | **DETR** (Zhu et al. (2020)) | - | - | - | **41.0** | **55.7** | **430.10** |
| ANNs to SNNs | Vthfast+Vthacc (Kim et al. (2020a)) | 8.7 | 46.7 | - | 8.9 | 21.1 | - |
| | Spiking-YOLO (Kim et al. (2020b)) | 8.7 | 51.8 | - | 8.9 | 25.7 | - |
| | SpiCalib (Li et al. (2022b)) | 23.5 | 75.2 | - | 23.9 | 45.4 | - |
| | **SUHD (Qu et al. (2024))** | - | **75.3** | **-** | **-** | **54.6** | **-** |
| Dirctly trained SNNs | Hybrid-YOLO (Guo et al. (2024)) | - | **54.2** | 13.48 † | - | 26.2 | - |
| | EMS-YOLO (Su et al. (2023)) | - | - | - | 26.9 | 50.1 | 29.00 |
| | **SpikeFormer (Yao et al. (2024a))** | - | - | - | **75.0** | **51.2** | **140.80** |
| | **SpikSSD-S** | **14.5** | **74.1** | **7.11** | **16.3** | **49.2** | **11.65** |
| | **SpikSSD-L** | **19.5** | **76.0 (76.3‡)** | **10.16 (11.56‡)** | **21.3** | **52.4** | **15.59** |

Table 3: **Comparison with state-of-the-art models on the VOC 2007 test and COCO 2017 validation datasets.** † indicates that energy is recalculated using our energy consumption method for fair comparison. ‡ denotes results from models pre-trained on COCO and fine-tuned on VOC.

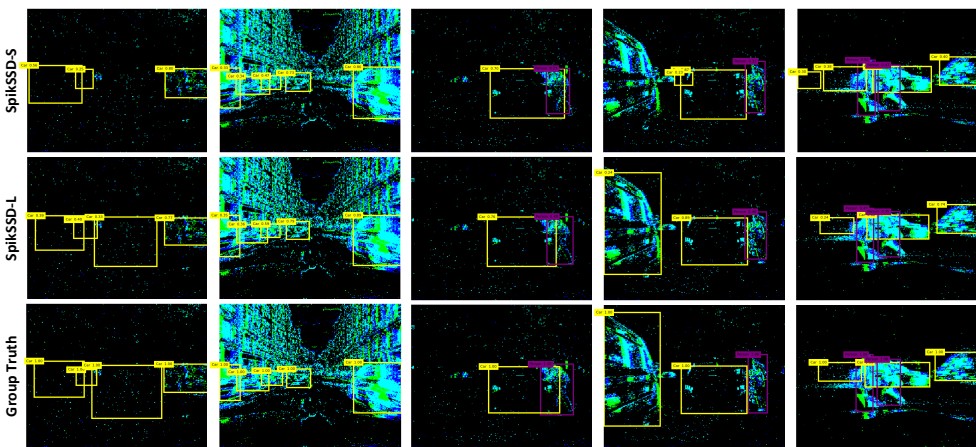

Figure 5: **Inference results on the GEN1 dataset.**

methods, SpikSSD-L not only demonstrates comparable performance but also requires only 1/3 (YOLOV5s) or even 1/37 (DETR) of the energy consumption. To the best of our knowledge, since Hybrid-YOLO is a hybrid model, SpikSSD is the first SNN model to demonstrate performance on the VOC 2007 dataset through direct training.

## 5 CONCLUSION

In this paper, we propose a novel and efficient SNN-based object detector, SpikSSD. Specifically, to enhance the model feature extraction capability, we introduce MDS-ResNet, which stabilizes membrane synaptic input through MDS, achieving optimal performance with the most efficient firing pattern. For feature fusion, we introduce the first SNN-based bi-directional fusion module, the SBFM. Experimental results show that our SpikSSD achieves state-of-the-art performance, surpassing other SNN-based models on the GEN1, VOC 2007, and COCO 2017 datasets. Notably, on the GEN1 dataset, this is the first full-spiking SNN model to achieve an mAP exceeding 40.0, demonstrating superior feature extraction and fusion capabilities. In the future, we will focus on optimizing detection heads for SNN-based object detectors, an unexplored area in both our research and SNN-specific designs.

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

# A  APPENDIX

## A.1  DISCUSSION

### A.1.1  DEPLOYMENT ON NEUROMORPHIC CHIP

In this paper, we have conducted theoretical energy evaluations to compare energy consumption between ANNs and SNNs, following common practices in SNN research (Fan et al. (2024); Su et al. (2023)). Our method shows substantial energy advantages compared to both SNN and ANN solutions. Detailed evaluation methods are provided in Appendix A.3.2.

It's worth noting that the above theoretical analysis excludes hardware architecture or data caching. The efficiency gains may be further amplified when SNNs are deployed on neuromorphic chips. These neuromorphic platforms, such as the Speck chip (Yao et al. (2024b)), operate in asynchronous mode without a global clock, leveraging their spike-driven characteristics. This design leads to minimal static energy consumption during periods without spike inputs. Consequently, the overall energy consumption of neuromorphic computing primarily scales with the number of spike-induced ACs (Pei et al. (2019)).

For deployment to real hardware, we take the neuromorphic chip Speck as an example. First, we need to use Sinabs, a PyTorch-based deep learning library, to build and train SNN models. Through the Sinabs-Speck plugin, we can complete model quantization and configuration generation. Subsequently, we can use the Samna tool for deployment and execution.

### A.1.2  THE LEARNING CURVE WITH AND WITHOUT SBFM

In this paper, our proposed SBFM demonstrates advantages in accelerating model convergence and stabilizing the training process, as shown in Fig. 6. We attribute this improvement to SBFM's construction of two shorter backpropagation paths from the deepest features to the input, significantly benefiting training and convergence. Therefore, SBFM not only enhances performance but also provides these additional advantages.

## A.2  PROOF OF BLOCK DYNAMICAL ISOMETRY

**Lemma 3. (Addition)** Chen et al. (2020) *Given* $\mathbf{J} := \sum_j \mathbf{J_j}$, *where* $\mathbf{J_j}$ *is a series of independent random matrices. If at most one matrix in* $\mathbf{J_j}$ *is not a central matrix, we have*

$$\phi\left(\mathbf{J}\mathbf{J}^T\right) = \sum_j \phi\left(\mathbf{J_j}\mathbf{J_j}^T\right) \tag{9}$$

**Proposition 1.** *For MDS-Block1, MDS-Block2, and MDS-Block3, when the block output follows* $\mathbf{x} \sim \mathcal{N}(0, 1)$, *each satisfies:* $\phi(\mathbf{J_j}\mathbf{J_j^T}) \approx \frac{1}{\alpha_2^{j-1}}$.

**Proof of MDS-Block1.** The MDS-Blocks consist of two distinct pathways: the residual path and the shortcut path. Consequently, we designate their respective Jacobian matrices as $\mathbf{J}_{\text{res}}$ and $\mathbf{J}_{\text{sc}}$. For brevity, we omit the layer number l where it doesn't cause ambiguity.

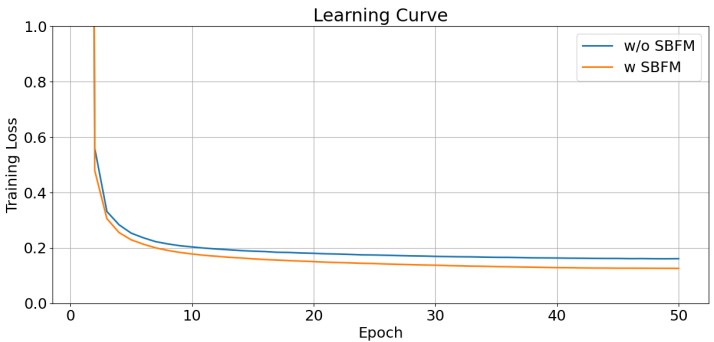

Figure 6: **The Learning Curve with and without SBFM.**

For the shortcut path, as proven in Su et al. (2023), it satisfies:

$$\phi(\mathbf{J_{sc}J_{sc}}^T) = \frac{\alpha_2^{maxpool}}{\alpha_2^{l-1}}(\frac{c_{j-1}}{c_j} + \frac{\delta_j}{c_j}\phi\left(\mathbf{H_j H_j}^T\right))$$

$$= \frac{1}{\alpha_2^{l-1}}(\frac{\alpha_2^{maxpool}c_{j-1}}{c_j} + \frac{\alpha_2^{maxpool}\delta_j}{c_j}\left(\frac{\alpha_2^{bn}}{\alpha_2^{maxpool}}\right)) \tag{10}$$

The Jacobian matrix of the LCB block Su et al. (2023) is represented by $\mathbf{H_j}$. The input and output channels for concatenation are represented by $c_{j-1}$ and $c_j$ respectively. Additionally, $\delta_j$ represents the difference between $c_j$ and $c_{j-1}$. Since $\alpha_2^{l-1}$ and $\alpha_2^{maxpool}$ can be set as fixed values, we only need to initialize the BN layer such that $\alpha_2^{bn} = \frac{2c_j - \alpha_2^{maxpool}c_{j-1}}{\delta_j}$. This initialization ensures that $\phi(\mathbf{J_{sc}J_{sc}}^T) = \frac{1}{\alpha_2^{l-1}}$.

For the residual path, since all its components satisfy Definition 1, we can derive:

$$\alpha_2^{l,res} = \phi(\mathbf{J_{res}J_{res}}^T)\alpha_2^{l-1} \tag{11}$$

$$\phi(\mathbf{J_{res}J_{res}}^T) = \frac{\alpha_2^{l,res}}{\alpha_2^{l-1}} \tag{12}$$

According to Lemma 3, we can derive:

$$\phi(\mathbf{J_{MDS-Block1}J_{MDS-Block1}}^T) = \phi(\mathbf{J_{sc}J_{sc}}^T) + \phi(\mathbf{J_{res}J_{res}}^T)$$

$$= \frac{1 + \alpha_2^{l,res}}{\alpha_2^{l-1}} \tag{13}$$

In residual learning, the goal is to have the residual path learn a small perturbation relative to the shortcut path to optimize the model's feature representation. To achieve this, as in Su et al. (2023); Hu et al. (2024), we initialize the tdBN of the residual path with a mean of 0 and a variance close to 0, avoiding loss of learning ability due to 0 variance. Correspondingly, we initialize the tdBN of the shortcut path with a mean of 0 and a variance of 1. Consequently, at this point, the output distribution of the block is approximately equal to the output of the shortcut, following $\mathbf{x} \sim \mathcal{N}(0,1)$. Therefore, their firing patterns are also approximately the same, achieving the goal of learning a small perturbation. This implies that $\alpha_2^{l,res} \approx 0$. As a result, the MDS-Block1 satisfies: $\phi(\mathbf{J_{MDS-Block1}J_{MDS-Block1}}^T) \approx \frac{1}{\alpha_2^{j-1}}$.

**Proof of MDS-Block2.** For the shortcut path in MDS-Block2, each of its components satisfies Definition 1. Therefore, we can draw a conclusion similar to that of the residual path in MDS-Block1:

$$\phi(\mathbf{J_{sc}J_{sc}}^T) = \frac{\alpha_2^{l,sc}}{\alpha_2^{l-1}} \tag{14}$$

Regarding the residual path in MDS-Block2, its structure is identical to that in MDS-Block1, so we don't elaborate on it here. According to Lemma 3, we can derive:

$$\phi(\mathbf{J}_{\mathbf{MDS-Block2}}\mathbf{J}_{\mathbf{MDS-Block2}}{}^T) = \frac{\alpha_2^{l,sc} + \alpha_2^{l,res}}{\alpha_2^{l-1}} \tag{15}$$

According to the proof of MDS-Block1, similarly, as long as we ensure that the output of the block follows $\mathbf{x} \sim \mathcal{N}(0,1)$, the MDS-Block2 satisfies: $\phi(\mathbf{J}_{\mathbf{MDS-Block2}}\mathbf{J}_{\mathbf{MDS-Block2}}{}^T) \approx \frac{1}{\alpha_2^{j-1}}$.

**Proof of MDS-Block3.** For the shortcut path in MDS-Block3, each of its components satisfies Definition 1. Moreover, the residual path of MDS-Block3 is identical to that of MDS-Block1 and MDS-Block2. Therefore, we can also conclude similar to that of MDS-Block2:

$$\phi(\mathbf{J}_{\mathbf{MDS-Block3}}\mathbf{J}_{\mathbf{MDS-Block3}}{}^T) = \frac{\alpha_2^{l,sc} + \alpha_2^{l,res}}{\alpha_2^{l-1}} \tag{16}$$

Therefore, with the output of the block following the distribution $\mathbf{x} \sim \mathcal{N}(0,1)$, the MDS-Block2 satisfies: $\phi(\mathbf{J}_{\mathbf{MDS-Block2}}\mathbf{J}_{\mathbf{MDS-Block2}}{}^T) \approx \frac{1}{\alpha_2^{j-1}}$.

**Proposition 2.** *For MDS-ResNet, when the encoding layer output follows $\mathbf{x} \sim \mathcal{N}(0, 2^{N_{\text{MDS-Block4}}})$ and each block output follows $\mathbf{x} \sim \mathcal{N}(0,1)$, the network satisfies: $\phi(\mathbf{J}\mathbf{J}^T) \approx 1$.*

For MDS-Block4, its structure is consistent with MS-Block Hu et al. (2024). Therefore, according to the results in Su et al. (2023), we can derive:

$$\phi(\mathbf{J}_{\mathbf{MDS-Block4}}\mathbf{J}_{\mathbf{MDS-Block4}}{}^T) = \frac{\alpha_2^{j-1} + 1}{\alpha_2^{j-1}} \tag{17}$$

At this point, if the output of each block follows $\mathbf{x} \sim \mathcal{N}(0,1)$, then $\alpha_2^{j-1} = 1$. Thus, we can conclude that MDS-Block4 satisfies: $\phi(\mathbf{J}_{\mathbf{MDS-Block4}}\mathbf{J}_{\mathbf{MDS-Block4}}{}^T) = 2$.

According to Lemma 2, based on the properties of individual blocks, the MDS-ResNet satisfies:

$$\phi(\mathbf{J}\mathbf{J}^T) = \frac{2^{N_{\text{MDS-Block4}}}}{\alpha_2^0} \tag{18}$$

Therefore, it suffices to maintain the condition that each block follows $\mathbf{x} \sim \mathcal{N}(0,1)$ and to establish $\alpha_2^0 = 2^{N_{\text{MDS-Block4}}}$. Under these conditions, the following holds: $\phi(\mathbf{J}\mathbf{J}^T) \approx 1$.

## A.3 MORE EXPERIMENTAL SETUP

### A.3.1 DATASETS INTRODUCTION

The GEN1 dataset (De Tournemire et al. (2020)) represents the initial large-scale collection for object detection using event cameras. It comprises car footage spanning over 39 hours, captured by the GEN1 device with a spatial resolution of 304x240. The dataset includes bounding box annotations for vehicles and pedestrians, provided at rates of 1 to 4Hz. These labels accumulate to a total exceeding 255,000 across the recordings.

The PASCAL VOC dataset (Everingham et al. (2010)) is a popular large-scale object detection dataset. It is divided into two subsets: VOC 2007 and VOC 2012. As the test set of VOC 2012 is not publicly available, we use VOC 2007+2012 trainval for training and VOC 2007 test for testing. The VOC 2007+2012 trainval contains a total of 16,551 images and 40,058 objects, while the VOC 2007 test comprises 4,952 images and 12,032 objects.

The COCO 2017 (Lin et al. (2014)) is a large-scale object detection benchmark with 118,287 training and 5000 validation images. Objects from 80 categories are annotated with their classes and locations.

| Stage | Output Size | MDS-ResNet10 | MDS-ResNet18 | MDS-ResNet34 |
|---|---|---|---|---|
| Conv1 | 1/2 | 7×7, 32, stride 2 | | |
| Conv2_x | 1/4 | $\begin{bmatrix} 3 \times 3, 32 \\ 3 \times 3, 64 \end{bmatrix} \times 1$ | $\begin{bmatrix} 3 \times 3, 32 \\ 3 \times 3, 64 \end{bmatrix} \times 2$ | $\begin{bmatrix} 3 \times 3, 32 \\ 3 \times 3, 64 \end{bmatrix} \times 3$ |
| Conv3_x | 1/8 | $\begin{bmatrix} 3 \times 3, 64 \\ 3 \times 3, 128 \end{bmatrix} \times 1$ | $\begin{bmatrix} 3 \times 3, 64 \\ 3 \times 3, 128 \end{bmatrix} \times 2$ | $\begin{bmatrix} 3 \times 3, 64 \\ 3 \times 3, 128 \end{bmatrix} \times 4$ |
| Conv4_x | 1/16 | $\begin{bmatrix} 3 \times 3, 128 \\ 3 \times 3, 256 \end{bmatrix} \times 1$ | $\begin{bmatrix} 3 \times 3, 128 \\ 3 \times 3, 256 \end{bmatrix} \times 2$ | $\begin{bmatrix} 3 \times 3, 128 \\ 3 \times 3, 256 \end{bmatrix} \times 6$ |
| Conv5_x | 1/32 | $\begin{bmatrix} 3 \times 3, 256 \\ 3 \times 3, 512 \end{bmatrix} \times 1$ | $\begin{bmatrix} 3 \times 3, 256 \\ 3 \times 3, 512 \end{bmatrix} \times 2$ | $\begin{bmatrix} 3 \times 3, 256 \\ 3 \times 3, 512 \end{bmatrix} \times 3$ |

Table 4: **The Structure of MDS-ResNet.** The x indicates that the corresponding block is repeated x times.

### A.3.2 ENERGY CONSUMPTION

The energy efficiency of SNNs primarily comes from performing accumulation calculations (AC) only when neurons fire. However, many current SNN-based works cannot guarantee a full-spiking network. Therefore, when calculating the energy consumption of these networks, we also consider multiplication and addition (MAC) operations. For ANNs, as the vast majority of their operations are MAC computations with only a small number of AC operations, we ignore these AC operations in our calculations. Furthermore, in line with previous works Fan et al. (2024); Su et al. (2023), we quantify the specific energy consumption values using $E_{\text{MAC}} = 4.6\text{pJ}$ and $E_{\text{AC}} = 0.9\text{pJ}$. The energy consumption calculation formulas for SNNs and ANNs are shown respectively as follows, where fr represents the firing rate of the model, and T denotes the number of time steps.

$$E_{SNNs} = T \times fr \times (E_{AC} \times N_{AC} + E_{MAC} \times N_{MAC}) \tag{19}$$

$$E_{ANNs} = T \times E_{MAC} \times N_{MAC} \tag{20}$$

### A.3.3 MORE IMPLEMENTATION DETAILS

The MDS-ResNet structure is illustrated in Tab. 4. Notably, we employ MDS-Block1 for downsampling in the first block of each stage. In MDS-ResNet18, given the shallower depth of the model, gradient propagation is not a concern. Therefore, we opt for MDS-Block3 when no downsampling is required, without using MDS-Block4. In MDS-ResNet34, we utilize MDS-Block3 in the middle block of each stage to enhance the stability of membrane synaptic input distribution. The remaining blocks implement MDS-Block4 to facilitate gradient propagation. Furthermore, for the object detection task in this study, we incorporate two additional downsampling stages, Conv6 and Conv7, both composed exclusively of MDS-Block2.

To ensure the network's firing rate, we set the threshold $V_{th}$ of the LIF/PLIF neurons to 1, to maintain the balance between the membrane synaptic input and the neuron firing threshold (Zheng et al. (2021)). On the Gen1 dataset, the model is trained for 50 epochs with batch size 32. However, we observe that MDS-ResNet34 fails to converge after 50 epochs on this dataset. Therefore, we extend the training to 100 epochs for all models using MDS-ResNet34 as the backbone. For PASCAL VOC, we train for 400 epochs with batch size 32. For COCO 2017, we train for 300 epochs with batch size 24.

To address the inefficiency in non-overlapping cases and aspect ratio insensitivity of IoU-based losses, we employ the Complete IoU (CIOU) Loss (Zheng et al. (2020)). CIOU Loss optimizes overlap area, central point distance, and aspect ratio simultaneously, enhancing accuracy and convergence in object detection. Furthermore, to address the disparity in class distribution between

| Backbone | Neuron | Fusion | Fusion Layers | mAP | mAP @0.5 | Param (M) | Firing Rate(%) | Energy (mJ) |
|---|---|---|---|---|---|---|---|---|
| MDS-ResNet34† | LIF | None | 0 | 35.7 | 62.1 | 14.8 | 17.26 | 2.34 |
| MDS-ResNet34‡ | LIF | None | 0 | 35.8 | 62.3 | 15.7 | 11.83 | 1.74 |
| **MDS-ResNet34** | **LIF** | **None** | **0** | **37.4** | **64.1** | **15.2** | **12.41** | **1.73** |
| MDS-ResNet18 | LIF | SFM | 5 | 36.2 | 63.4 | 12.6 | 16.49 | 8.04 |
| MDS-ResNet18 | LIF | SSFM | 5 | 36.5 | 63.2 | 12.5 | 11.01 | 1.43 |
| MDS-ResNet18* | LIF | SBFM | 5 | 36.9 | 63.6 | 12.1 | 11.32 | 1.25 |
| MDS-ResNet18 | PLIF | SBFM | 5 | 37.5 | 63.9 | 13.9 | 10.39 | 1.44 |
| MDS-ResNet18 | LIF | SBFM | 3 | 33.9 | 59.9 | 8.2 | 10.06 | 1.33 |
| MDS-ResNet18 | LIF | SBFM | 4 | 36.2 | 62.7 | 12.2 | 10.70 | 1.48 |
| **MDS-ResNet18** | **LIF** | **SBFM** | **5** | **37.6** | **64.3** | **13.9** | **10.57** | **1.47** |
| MDS-ResNet18 | LIF | SBFM | 6 | 38.4 | 65.6 | 15.4 | 10.76 | 1.50 |

Table 5: **Results of the more ablation studies on the GEN1 dataset.** A * indicates that Spiking Depthwise Separable Convolution is not used in SBFM. A † indicates that only MDS-Block4 is used, and ‡ indicates that only MDS-Block3 is used when the model does not downsample.

| Backbone | Neuron | Input size | mAP | mAP @0.5 | Param (M) | Firing Rate(%) | Energy (mJ) |
|---|---|---|---|---|---|---|---|
| MDS-ResNet34* | LIF | 512 | 45.8 | 73.3 | 19.5 | 13.83 | 9.90 |
| MDS-ResNet34* | PLIF | 512 | 46.3 | 73.8 | 19.5 | 13.77 | 9.85 |
| MDS-ResNet34 | PLIF | 300 | 45.4 | 72.4 | 19.5 | 13.55 | 3.48 |
| **MDS-ResNet34** | **PLIF** | **512** | **48.9** | **76.0** | **19.5** | **14.19** | **10.16** |
| MDS-ResNet34 | PLIF | 576 | 47.6 | 75.0 | 19.5 | 13.88 | 12.58 |
| MDS-ResNet34 | PLIF | 640 | 48.8 | 76.0 | 19.5 | 14.19 | 15.88 |
| MDS-ResNet34 | PLIF | 800 | 48.1 | 73.6 | 19.5 | 13.90 | 24.36 |

Table 6: **Results of the more ablation studies on the VOC 2007 dataset.** A * indicates that the model is trained for 300 epochs.

objects and non-objects during detector training, we adopt Focal Loss (Lin et al. (2017)) as our classification criterion.

The formula for CIOU Loss is shown in Equation 21, where IoU is the intersection over union, $\rho^2(b, b^g t)$ is the squared central point distance, $c^2$ is the enclosing box's diagonal length squared, and $\alpha v$ addresses aspect ratio consistency. The Focal Loss formula is shown in Equation 22, where $p_t$ is the model's estimated probability for the true class, $\alpha_t$ is a balancing factor for class frequency, and $\gamma$ is the focusing parameter that reduces the loss contribution from easy examples.

$$\text{CIOU} = 1 - \text{IoU} + \frac{\rho^2(b, b^{gt})}{c^2} + \alpha v \tag{21}$$

$$FL(p_t) = -\alpha_t(1 - p_t)^\gamma \log(p_t) \tag{22}$$

## A.4 MORE ABLATION STUDIES

In this section, we present additional ablation studies.

### A.4.1 THE COMBINATION OF MDS-BLOCK3 AND MDS-BLOCK4

To validate the effectiveness of combining MDS-Block3 and MDS-Block4, we conduct experiments as shown in rows 1 to 3 of Tab. 5. In deeper networks, using MDS-Block3 alone only marginally improves mAP compared to MDS-ResNet18, due to the gradient degradation problem. Furthermore, as shown in row 1, employing only MDS-Block4 also results in performance degradation, which validates the effectiveness of our proposed MDS. Consequently, the integration of both blocks proves to be both necessary and effective.

### A.4.2    Bi-direction Fusion versus One-way Fusion

In Section 3.4, we qualitatively analyze the superiority of bi-directional fusion over one-way fusion. We quantitatively demonstrate this point in rows 4, 5 and 10 of Tab. 5. To facilitate this comparison, we create two variants: replacing SBFM with SFM (Fan et al. (2024)), and removing the up-down fusion process from SBFM to create a Spiking Single Fusion Module (SSFM) that performs one-way fusion. We then compare their performance against the complete SBFM. As observed, the model employing SBFM achieves a significant improvement in mAP while consuming almost the same or even less energy. This evidence supports our claim that the SBFM with a bi-directional fusion strategy enables more comprehensive feature integration.

### A.4.3    The Effectiveness of Spiking Depthwise Separable Convolution

In Section 3.4, to enhance fusion in both channel and spatial dimensions, we replace the MDS in MDS-Block3 with spiking depthwise separable convolution to form MDSF-Block. As shown in rows 6 and 10 of Tab. 5, experimental results demonstrate that this substitution significantly improves performance. We attribute this improvement to the addition of spiking depthwise convolution in the spiking depthwise separable convolution, which, despite increasing the parameters, enhances spatial feature extraction capability compared to the original MDS, thereby boosting our model performance.

### A.4.4    The Number of Fusion Layers in SBFM

We conduct a quantitative investigation into the number of fusion layers in SBFM. As shown in rows 8-11 of Tab. 5, increasing the number of fusion layers from 3 to 6 leads to significant performance improvements. We attribute this to the fact that SNNs inherently have limited feature representation capability due to their spike-based computation nature, and incorporating multiple-scale features helps enhance multi-scale detection performance. While increasing the number of fusion layers to 6 exhibits the highest mAP, we observe that the gains are marginal compared to the additional parameters and energy consumption introduced. Therefore, we determine that using 5 fusion layers achieves the optimal balance between performance and computational efficiency.

### A.4.5    The LIF Neuron versus PLIF Neuron

We compare the two most commonly used spiking neuron models (LIF and PLIF) on both the event-based Gen1 dataset and the frame-based VOC 2007 dataset. As shown in rows 7 and 10 of Tab. 5 and rows 1 and 2 of Tab. 6, LIF neurons outperform PLIF neurons on the event-based dataset, while PLIF neurons excel on the frame-based dataset. We attribute this difference to PLIF neurons providing the model with greater expressive power. This becomes particularly advantageous for SNNs when processing frame-based datasets, which are inherently more complex and challenging for SNNs compared to event-based datasets, as the latter's sparse nature makes them naturally suited for SNN processing.

### A.4.6    Infulence of Input Size on Pascal VOC

We also analyze the influence of input size on performance for the frame-based Pascal VOC dataset. As shown in rows 3-7 of Tab. 6, similar to the results on the GEN1 dataset, the model's mAP@0.5 increases when the input scale is enlarged to 512. However, further increasing the input size leads to a decline in mAP@0.5. This phenomenon might be attributed to excessively large input sizes reducing the number of small and medium-sized objects in the input, thereby diminishing the multi-scale detection capabilities of the model.

## A.5    More Visualization

In this section, we present the visualization results of our model on the PASCAL VOC dataset, as shown in Fig. 7. These results demonstrate that our model can achieve excellent performance on frame-based datasets. Moreover, we observe that the effectiveness of model improves significantly as its depth increases.

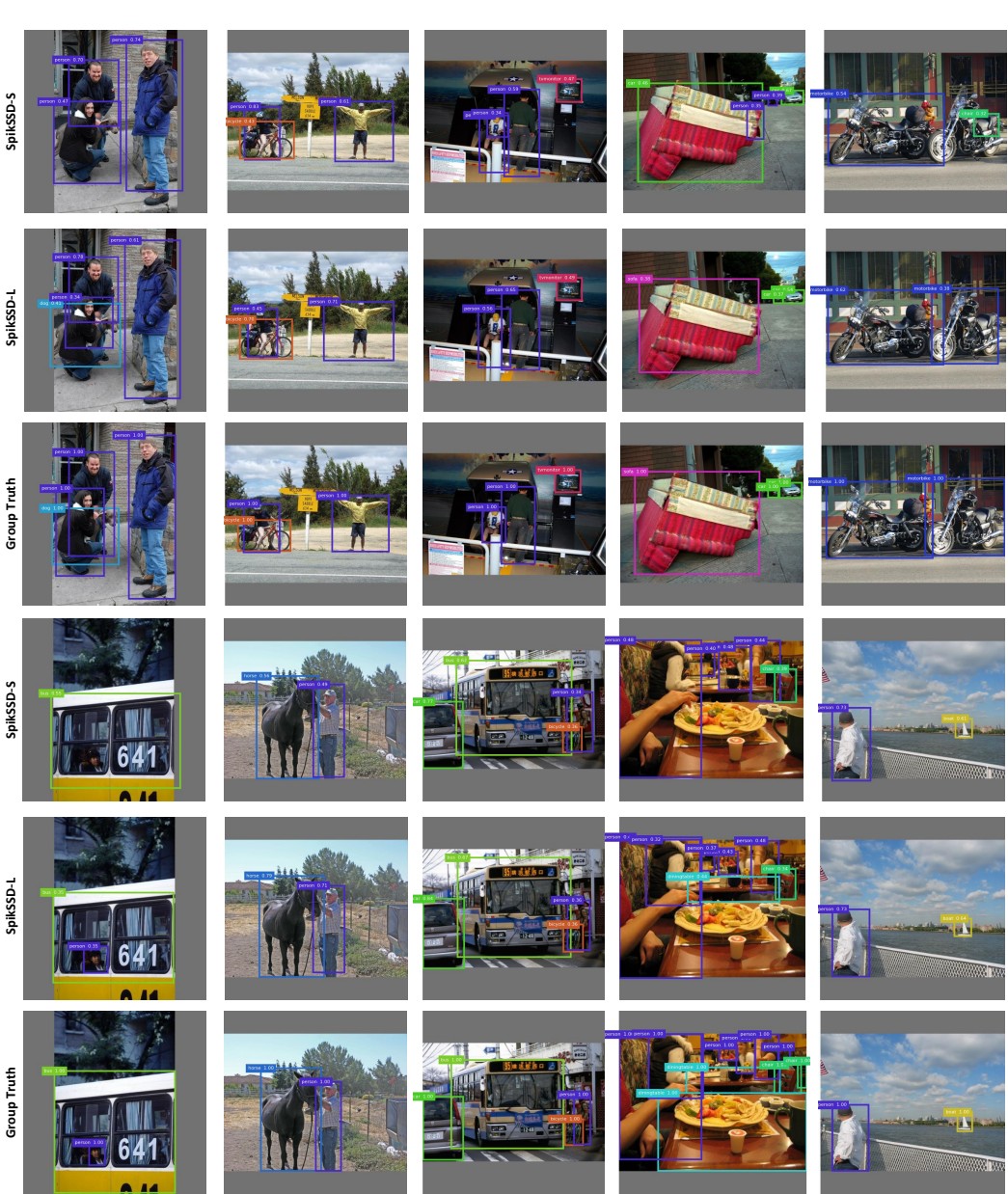

Figure 7: **Inference results on the PASCAL VOC dataset.**

