# OpenReview forum: "SpikSSD: Better Extraction and Fusion for Object Detection with Spiking Neuron Networks"
_ICLR.cc/2025/Conference — Submitted to ICLR 2025_

### Official Review · Reviewer_V2pm · 2024-10-28

**Soundness:** 3
**Presentation:** 2
**Contribution:** 2
**Rating:** 3
**Confidence:** 5

**Summary:**

This paper proposes a spiking neural network backbone MDS-ResNet, and a spiking object detection model SpikSSD. Based on EMS-ResNet, the proposed MDS-ResNet modifies the shortcut connection in the residual blocks. The SpikSSD combines the proposed bi-directional feature fusion and MDS-ResNet. Experimental results show that the proposed SpikSSD outperforms existing spiking object detection methods.

**Strengths:**

1. This paper analyzes in detail the gradient vanishing or explosion for MDS-ResNet.
2. The proposed SpikSSD model achieves state-of-the-art mAP metric performance on GEN1 and VOC2007 datasets compared to existing spiking object detection methods.

**Weaknesses:**

1. In the method section, the authors spend a lot of words introducing the proposed backbone MDS-ResNet. However, MDS-Blocks 1, 2, and 4 are already utilized in MS-ResNet and EMS-ResNet, while only MDS-Block 3 is newly proposed, and it closely resembles MDS-Block 2. Moreover, the gradient analysis of MDS-ResNet is similar to EMS-ResNet. Therefore, I believe that the MDS-ResNet architecture should not be regarded as a main contribution of this paper. I suggest that the authors detail why MDS-Block is more effective for object detection tasks than MS-Block and EMS-Block.
2. The SpikSSD is only evaluated on VOC 2007 and GEN1 datasets, while EMS-ResNet is evaluated mainly on COCO 2017 dataset. I wonder if SpikSSD also works well on COCO 2017. If possible, please provide the experimental results on COCO 2017 dataset.
3. Figure 1 illustrates the firing patterns of MDS-ResNet and EMS-ResNet. However, it does not indicate on which task or dataset the evaluation is performed. Furthermore, it is not clear whether this phenomenon holds for all tasks or only for specific tasks.

**Questions:**

Please refer to the weaknesses.

---

> ### Author Response · Authors · 2024-11-23
> **About Questions (Part 1/2)**
>
> Thanks for your thoughtful review. We are pleased that you have recognized our superior performance on GEN1 and VOC datasets for SNN-based object detection. We would like to address your concerns and questions in the following sections.
>
> Q1: In the method section, the authors spend a lot of words introducing the proposed backbone MDS-ResNet. However, MDS-Blocks 1, 2, and 4 are already utilized in MS-ResNet and EMS-ResNet, while only MDS-Block 3 is newly proposed, and it closely resembles MDS-Block 2. Moreover, the gradient analysis of MDS-ResNet is similar to EMS-ResNet. Therefore, I believe that the MDS-ResNet architecture should not be regarded as a main contribution of this paper. I suggest that the authors detail why MDS-Block is more effective for object detection tasks than MS-Block and EMS-Block.
>
> A: We appreciate the reviewer's thoughts on MDS-ResNet. In traditional membrane-based shortcuts [1], the output of the residual path continuously accumulates to the shortcut path. Assuming these outputs are approximately independent, the variance of their sum equals the sum of their variances, leading to a gradual increase in membrane synaptic input variance for subsequent neurons. This results in a higher probability of extreme inputs. Specifically, when the synaptic input is small, the accumulated membrane potential fails to reach the firing threshold. When it is excessively large, neurons exhibit abnormally high firing rates. In both cases, the firing pattern becomes either overly sparse or excessive, failing to reflect the input features accurately. This unstable firing pattern makes it difficult for the network to distinguish and extract meaningful features, which in turn affects the network’s feature extraction capability.
>
> To address this issue, we propose the Membrane-based Deformed Shortcut (MDS), which improves the shortcut by enabling it to adjust the distribution of the shortcut output. Based on MDS, we introduce MDS-Block3.
> During downsampling, since the shortcut in Spiking Residual Networks already incorporates convolution and BatchNorm operations, this is not our primary focus for improvement.
> Therefore, for downsampling operations, we adopt the idea from EMS-ResNet, utilizing MDS-Block1 and MDS-Block2.
> Notably, as the model goes deeper, we also retain the original MS-Block as MDS-Block4 to facilitate gradient propagation. This combination of the MDS-Block3 and MDS-Block4 prevents degradation of gradient flow in deeper networks, which could occur if only MDS-Block3 is used throughout the model. We have conducted experiments on MDS-ResNet34 to validate this design choice, and the results shown in rows 4-6 of Tab. R5 strongly support the effectiveness of combining these two blocks.
>
> This paper aims to stabilize membrane synaptic input through MDS, a simple yet effective approach, thereby achieving optimal performance with the most efficient firing pattern. Extensive experiments, as shown in rows 1-3 of Tab. R5, demonstrate the superior performance of our proposed MDS-ResNet compared to MS-ResNet and EMS-ResNet. Therefore, we think it is appropriate to consider the introduction of MDS-ResNet as a contribution of this work.
>
> | Backbone         | Param(M) |   mAP    | mAP@0.5  | Firing Rate(%) | Energy(mJ) |
> | :--------------- | :------: | :------: | :------: | :------------: | :--------: |
> | EDS-ResNet18     |   9.7    |   32.9   |   59.2   |     17.51      |    1.29    |
> | MS-ResNet18      |   9.8    |   34.1   |   60.4   |     12.80      |    2.04    |
> | **MDS-ResNet18** | **10.1** | **34.7** | **60.9** |   **12.41**    |  **0.97**  |
> | MDS-ResNet34†    |   14.8   |   35.7   |   62.1   |     17.26      |    2.34    |
> | MDS-ResNet34‡    |   15.7   |   35.8   |   62.3   |     11.83      |    1.74    |
> | **MDS-ResNet34** | **15.2** | **37.4** | **64.1** |   **12.41**    |  **1.73**  |
>
> **Table R5: Performance comparison between different spiking backbones and model configurations on the GEN1 dataset.** A † indicates that only MDS-Block4 is used, and ‡ indicates that only MDS-Block3 is used when the model does not downsample.
>
> Furthermore, since both our work and EMS-ResNet employ the Block Dynamical Isometry [2] method for gradient analysis, the analytical approach should naturally be similar. Moreover, the purpose of conducting gradient analysis on MDS-ResNet is to demonstrate that the MDS does not affect the training and convergence, rather than being an innovation point of this paper.
>
> Q2: The SpikSSD is only evaluated on VOC 2007 and GEN1 datasets, while EMS-ResNet is evaluated mainly on COCO 2017 dataset. I wonder if SpikSSD also works well on COCO 2017. If possible, please provide the experimental results on COCO 2017 dataset.
>
> A: Thanks for the suggestion. We have compared our method with other state-of-the-art approaches on the COCO 2017 dataset. Please refer to the section of To All Reviewers.

---

> > ### Author Response · Authors · 2024-11-23
> > **About Questions (Part 2/2)**
> >
> > Q3: Figure 1 illustrates the firing patterns of MDS-ResNet and EMS-ResNet. However, it does not indicate on which task or dataset the evaluation is performed. Furthermore, it is not clear whether this phenomenon holds for all tasks or only for specific tasks.
> >
> > A: We appreciate your thoughts and suggestions on Fig. 1. The firing patterns of both models shown in the figure are obtained from training on the GEN1 object detection dataset. We have added this to the paper. From the figure, it can be seen that our MDS effectively stabilizes the firing pattern in object detection tasks. This indicates that MDS-ResNet can reflect the input features accurately, thus demonstrating stronger feature extraction capability.
> >
> > Since the objective of proposing MDS-ResNet in this paper is to address the insufficient feature extraction capability of current spiking backbone networks in object detection, other tasks are beyond our current scope of discussion and research. Therefore, we have not investigated whether this phenomenon exists in other tasks. We appreciate your insightful suggestion and will investigate similar issues across different tasks in our future research.
> >
> > [1]  Hu Y, Deng L, Wu Y, et al. Advancing spiking neural networks toward deep residual learning[J]. IEEE Transactions on Neural Networks and Learning Systems, 2024.
> >
> > [2] Chen Z, Deng L, Wang B, et al. A comprehensive and modularized statistical framework for gradient norm equality in deep neural networks[J]. IEEE Transactions on Pattern Analysis and Machine Intelligence, 2020, 44(1): 13-31.

---

> > > ### Comment · Reviewer_V2pm · 2024-11-27
> > >
> > > Thank you for your reply. To the best of my knowledge, the motivation of this paper is like this: we propose MDS-Block3 (B3) to enhance the network’s feature extraction capability. However, B3 has some shortcomings, therefore we use B4 to overcome those shortcomings. Moreover, some modules have already used a similar mechanism, therefore we directly use B1 and B2.
> > >
> > > Such a motivation is too complex and not straightforward. It seems that the proposed B3 has a lot of problems, so you will have to add a lot of patches to fix it. These patches limit the contribution of the MDS-ResNet architecture.
> > >
> > > Moreover, it is still unclear why the proposed MDS-ResNet architecture is more effective for object detection tasks. If the proposed B3 only enhances the network’s feature extraction capability, it should also benefit other downstream tasks such as segmentation.
> > >
> > > In summary, I believe this paper would benefit from further revision. I recommend the authors clarify the motivation of this paper in future revisions. Therefore, I keep my original rating.

---

> > > > ### Author Response · Authors · 2024-11-27
> > > > **About Official Comment**
> > > >
> > > > Thanks for your thoughtful comments regarding MDS-ResNet. We want to clarify that our primary motivation for proposing MDS-ResNet is to address a critical issue in traditional membrane-based shortcuts: the continuous accumulation of residual path outputs in the shortcut path leads to increasing variance in membrane synaptic inputs for subsequent neurons. This results in a higher probability of extreme input, causing neuronal firing patterns to become either too sparse or excessive, which prevents accurate representation of features. This unstable firing pattern makes it difficult for the network to distinguish and extract meaningful features, which in turn affects the network’s feature extraction capability.
> > > >
> > > > | Backbone         | Param(M) |   mAP    | mAP@0.5  | Firing Rate(%) | Energy(mJ) |
> > > > | :--------------- | :------: | :------: | :------: | :------------: | :--------: |
> > > > | EDS-ResNet18     |   9.7    |   32.9   |   59.2   |     17.51      |    1.29    |
> > > > | MS-ResNet18      |   9.8    |   34.1   |   60.4   |     12.80      |    2.04    |
> > > > | **MDS-ResNet18** | **10.1** | **34.7** | **60.9** |   **12.41**    |  **0.97**  |
> > > > | MDS-ResNet34†    |   14.8   |   35.7   |   62.1   |     17.26      |    2.34    |
> > > > | MDS-ResNet34‡    |   15.7   |   35.8   |   62.3   |     11.83      |    1.74    |
> > > > | **MDS-ResNet34** | **15.2** | **37.4** | **64.1** |   **12.41**    |  **1.73**  |
> > > >
> > > > **Table R5: Performance comparison between different spiking backbones and model configurations on the GEN1 dataset.** A † indicates that only MDS-Block4 is used, and ‡ indicates that only MDS-Block3 is used when the model does not downsample.
> > > >
> > > > By incorporating our proposed MDS into B3 and combining it with other blocks, we introduce the MDS-ResNet that successfully addresses this issue. Our motivation is simple and straightforward. Both qualitative (Fig.1 in the paper) and quantitative (Tab. R5) experiments demonstrate that our method effectively resolves the aforementioned problem, validating our innovation. We believe the significance of this innovation should be evaluated based on the problems it solves, while the specific network architecture reflects a comprehensive consideration of various factors.
> > > > Specifically, in shallower networks like MDS-ResNet18, we omit B4. For deeper networks like MDS-ResNet34, using B3 exclusively would solve our initial motivation but lead to the degradation of gradient flow due to less clean shortcuts. By maintaining some B4 blocks, we achieve a natural balance that addresses both the original motivation and overcomes inherent limitations. This represents a simple trade-off rather than "add a lot of patches to fix it" issues.
> > > > Additionally, since downsampling is not our paper's focus, we naturally retain B1 and B2 without claiming them as innovations. Our primary goal is to solve problems through simple yet effective methods. The network's architectural trade-offs should not constrain our innovative contributions. We hope the reviewer will reconsider our paper and its innovations from this perspective.
> > > >
> > > > Furthermore, our paper's scope extends beyond merely improving feature extraction for detection tasks. We recognize that enhancing SNN performance in detection requires both efficient extraction and fusion of spiking features, as a strong backbone needs effective fusion to fully utilize its features, and vice versa. Therefore, beyond MDS-ResNet, we propose SBFM, which introduces bi-directional feature fusion to SNNs for the first time, enhancing multi-scale object detection capabilities. Integrating MDS-ResNet, SBFM, and the SSD Detection Head, our proposed SpikSSD achieves state-of-the-art mAP on event-based GEN1, frame-based VOC 2007, and COCO 2017 datasets compared to other SNN-based models, while maintaining the lowest firing rate.
> > > >
> > > > Finally, since the objective of proposing MDS-ResNet in this paper is to enhance object detection at the backbone network level, several prior works have also focused on improving detection tasks from this perspective without validating on other tasks [1, 2, 3, 4, 5]. However, we also recognize that MDS-ResNet enhances the feature extraction capability of spiking backbones, suggesting potential benefits for other applications as well. While this is beyond the scope of this study, we will explore these possibilities in our future work. We appreciate your interest and welcome you to follow our upcoming research.

---

> > > > > ### Author Response · Authors · 2024-11-27
> > > > > **References**
> > > > >
> > > > > [1] Su Q, Chou Y, Hu Y, et al. Deep directly-trained spiking neural networks for object detection[C]//Proceedings of the IEEE/CVF International Conference on Computer Vision. 2023: 6555-6565.
> > > > >
> > > > > [2] Liu Y, Wang Y, Wang S, et al. Cbnet: A novel composite backbone network architecture for object detection[C]//Proceedings of the AAAI conference on artificial intelligence. 2020, 34(07): 11653-11660.
> > > > >
> > > > > [3] Du X, Lin T Y, Jin P, et al. Spinenet: Learning scale-permuted backbone for recognition and localization[C]//Proceedings of the IEEE/CVF conference on computer vision and pattern recognition. 2020: 11592-11601.
> > > > >
> > > > > [4] Gehrig M, Scaramuzza D. Recurrent vision transformers for object detection with event cameras[C]//Proceedings of the IEEE/CVF conference on computer vision and pattern recognition. 2023: 13884-13893.
> > > > >
> > > > > [5] Chen Y, Yang T, Zhang X, et al. Detnas: Backbone search for object detection[J]. Advances in neural information processing systems, 2019, 32.

---

> > > > > > ### Author Response · Authors · 2024-12-01
> > > > > > **Official Comment by Authors**
> > > > > >
> > > > > > Dear Reviewer V2pm,
> > > > > >
> > > > > > We sincerely thank you for spending time and effort to review our manuscript and offer constructive feedback. As the discussion phase concludes, we hope our detailed responses have effectively addressed your concerns. If you need any additional clarifications or have unresolved questions, please feel free to contact us. We are more than willing to continue addressing your questions and enhancing our work based on your valuable insights.
> > > > > >
> > > > > > Best,
> > > > > >
> > > > > > Authors

---

### Official Review · Reviewer_TRoC · 2024-11-01

**Soundness:** 2
**Presentation:** 1
**Contribution:** 2
**Rating:** 3
**Confidence:** 5

**Summary:**

This paper introduces SpikSSD, a Spiking Single Shot Multibox Detector that leverages a fully spiking backbone network (MDS-ResNet) and a novel Spiking Bi-direction Fusion Module (SBFM) for object detection in Spiking Neural Networks (SNNs). The model shows strong performance on the GEN1 and PASCAL VOC 2007 datasets, achieving a mean average precision (mAP) comparable to or better than existing SNN-based models with a low firing rate.

**Strengths:**

The paper highlights the low energy consumption of SpikSSD, reporting approximately 10% firing rates, which is notable in comparison to conventional ANNs.

**Weaknesses:**

The results are primarily limited to GEN1 and VOC 2007, which may not fully represent the performance across diverse object detection tasks. The paper could be strengthened by including more diverse datasets or a comprehensive analysis of SpikSSD's generalization capabilities. While the paper introduces the MDS to stabilize membrane synaptic inputs, the theoretical foundations are only briefly discussed. A more rigorous exploration of why MDS effectively stabilizes membrane potential and prevents gradient vanishing/explosion would strengthen the paper’s contributions. Additionally, it’s unclear how the MDS compares with existing methods for managing synaptic input variance in SNNs. Although the results are promising, the comparisons are limited to SNN-based models. The paper would benefit from a broader comparison with non-SNN state-of-the-art object detectors on similar datasets to better highlight SpikSSD's competitiveness in real-world applications. Given that SNNs are often measured against ANN counterparts, direct ANN comparisons could showcase where SNNs might still be falling short. While SBFM achieves bi-directional fusion, the paper lacks a detailed justification of why this method is uniquely advantageous in SNNs as opposed to standard fusion techniques. It would also be helpful to clarify the specific benefits of using a spiking-only architecture over incorporating mixed approaches if they offer potential improvements in performance. The paper does not include ablation studies to validate the specific contributions of MDS-ResNet and SBFM. Without this, it is difficult to assess whether the performance gains are directly attributable to these components. Such studies could better clarify the impact of each proposed module.

**Questions:**

Certain sections lack clarity, particularly regarding the mechanisms of MDS and SBFM. Visualizations of neuron firing patterns and more illustrative figures could help elucidate these mechanisms. Additionally, providing more detail in the methodology could allow for better reproducibility.

---

> ### Author Response · Authors · 2024-11-23
> **About Weakness (Part 1/2)**
>
> Thanks for your thoughtful review. We are pleased that you have recognized our superior performance on GEN1 and VOC datasets for SNN-based object detection. We would like to address your concerns and questions in the following sections.
>
> Q1: The results are primarily limited to GEN1 and VOC 2007, which may not fully represent the performance across diverse object detection tasks. The paper could be strengthened by including more diverse datasets or a comprehensive analysis of SpikSSD's generalization capabilities.
>
> A: Thanks for the suggestion. We have compared our method with other state-of-the-art approaches on the COCO 2017 dataset. Please refer to the section of To All Reviewers.
>
> Q2: While the paper introduces the MDS to stabilize membrane synaptic inputs, the theoretical foundations are only briefly discussed. A more rigorous exploration of why MDS effectively stabilizes membrane potential and prevents gradient vanishing/explosion would strengthen the paper’s contributions.
>
> A: Thank you for your insightful comment regarding the theoretical foundations of MDS.
> We would like to clarify that our proposed MDS is to adjust the distribution of membrane synaptic input for subsequent neurons. Rather than stabilizing gradients, our analysis of gradient vanishing or explosion is conducted to demonstrate that the MDS does not affect the training and convergence.
>
> We would like to point out that comprehensive theoretical analyses of how MDS stabilizes membrane synaptic input distribution are already presented in our paper. Specifically, we provide the core theoretical framework in lines 70-85. This is further expanded with a detailed mathematical analysis in lines 206-274. Furthermore, to demonstrate that MDS does not affect training and convergence, we have provided extensive detailed mathematical analysis in lines 274-319, and Appendix (lines 743-846).
>
> To complement our theoretical analysis, we have conducted extensive ablation studies in Tab. 1 and provided visualization results in Fig. 1 of the paper. These empirical findings validate our theoretical claims and demonstrate the effectiveness of MDS in stabilizing membrane potential.
>
> Q3: Additionally, it’s unclear how the MDS compares with existing methods for managing synaptic input variance in SNNs.
>
> A: While existing methods for managing synaptic input variance in SNNs have primarily focused on extending BatchNorm to the temporal dimension [1, 2], the impact of traditional membrane-based shortcuts on synaptic input variance has been largely overlooked. Our analysis and experiments address this gap, providing novel insights that could benefit future SNN developments.
>
> Q4: Although the results are promising, the comparisons are limited to SNN-based models. The paper would benefit from a broader comparison with non-SNN state-of-the-art object detectors on similar datasets to better highlight SpikSSD's competitiveness in real-world applications. Given that SNNs are often measured against ANN counterparts, direct ANN comparisons could showcase where SNNs might still be falling short.
>
> A: As demonstrated in Tab. 2 and 3 of the paper, we have compared our work with non-SNN methods across GEN1 event camera datasets, COCO 2017, and VOC 2007 datasets. The results show that SpikSSD achieves comparable or even superior performance compared to ANN-based approaches.
>
> Q5: While SBFM achieves bi-directional fusion, the paper lacks a detailed justification of why this method is uniquely advantageous in SNNs as opposed to standard fusion techniques.
>
> A: We appreciate your thoughts on the proposed bi-directional fusion.
> In SNNs, features exist in both spatial and temporal domains. Spiking feature fusion enhances feature representation in both dimensions, improving multi-scale detection capabilities. This dual-domain nature makes feature fusion more crucial in SNNs compared to ANNs.
> Currently, the only fusion method designed for SNNs is the Spiking Fusion Module (SFM) [3]. This fusion method adopts a one-way, down-up approach, where large-scale features are upsampled to merge with small-scale features. Without up-down fusion, it only enhances small-scale feature representation, overlooking the potential improvement of large-scale features. Furthermore, SFM incorporates SEW-Block internally, which compromises the spiking nature of the network.
> To address these issues, we propose the Spiking Bi-direction Fusion Module (SBFM). Specifically, this module utilizes Spiking Up/Down Block and membrane addition-based fusion method to process feature maps, performing fusion first bottom-up and then top-down. We employ the proposed MDS Fusion Block (MDSF-Block) to extract fused features during this process. These SNN-friendly components ensure SBFM adheres to spiking characteristics while enhancing the model detection capabilities.
>
> We have incorporated this analysis into the paper.

---

> ### Author Response · Authors · 2024-11-23
> **About Weakness (Part 2/2)**
>
> Q6: It would also be helpful to clarify the specific benefits of using a spiking-only architecture over incorporating mixed approaches if they offer potential improvements in performance.
>
> A: We appreciate your thoughts on the spiking-only architecture. The spiking-only architecture better aligns with the inherent characteristics of SNNs, enabling more effective feature extraction. More importantly, such architecture can be readily deployed on neuromorphic chips like the Speck chip [4], fully leveraging the energy efficiency advantages of SNNs. In contrast, hybrid architectures, which typically combine spiking and traditional neural layers, present significant deployment challenges on neuromorphic hardware.
>
> Q7: The paper does not include ablation studies to validate the specific contributions of MDS-ResNet and SBFM. Without this, it is difficult to assess whether the performance gains are directly attributable to these components. Such studies could better clarify the impact of each proposed module.
>
> A: We have conducted comprehensive ablation studies in Tab. 1 of the paper to evaluate the individual contributions of MDS-ResNet and SBFM. The detailed analysis of these results can be found in Sections 4.2.1 (lines 412-420) and 4.2.3 (lines 428-450).

---

> > ### Author Response · Authors · 2024-11-23
> > **About Questions**
> >
> > Q1: Certain sections lack clarity, particularly regarding the mechanisms of MDS and SBFM. Visualizations of neuron firing patterns and more illustrative figures could help elucidate these mechanisms.
> >
> > A: We appreciate your thoughtful comments regarding the clarity of our paper. We would like to point out that comprehensive theoretical analyses of how MDS stabilizes membrane synaptic input are already presented in our paper. Specifically, we provide the core theoretical framework in lines 70-85. This is further expanded with a detailed mathematical analysis in lines 206-274. Furthermore, to demonstrate that MDS does not affect training and convergence, we have provided extensive detailed mathematical analysis in lines 274-319, and Appendix (lines 743-846).
> >
> > In addition, for SBFM, we have added corresponding descriptions, please refer to the response to Q5 in the About Weakness section.
> >
> > To complement our theoretical analysis, we have conducted extensive ablation studies in Tab. 1 and provided visualization results in Fig. 1 of the paper. These empirical findings validate our theoretical claims and demonstrate the effectiveness of MDS and SBFM.
> >
> > Q2: Additionally, providing more detail in the methodology could allow for better reproducibility.
> >
> > A: We appreciate your attention to the reproducibility of our work. We believe that the implementation details provided in Section 4.1 Experimental Setup and Appendix A.3.3 More Implementation Details are comprehensive. Furthermore, to ensure full reproducibility of our results, we have made our code publicly available in the supplementary materials.
> >
> > [1] Zheng H, Wu Y, Deng L, et al. Going deeper with directly-trained larger spiking neural networks[C]//Proceedings of the AAAI conference on artificial intelligence. 2021, 35(12): 11062-11070.
> >
> > [2] Duan C, Ding J, Chen S, et al. Temporal effective batch normalization in spiking neural networks[J]. Advances in Neural Information Processing Systems, 2022, 35: 34377-34390.
> >
> > [3] Fan Y, Zhang W, Liu C, et al. SFOD: Spiking Fusion Object Detector[C]//Proceedings of the IEEE/CVF Conference on Computer Vision and Pattern Recognition. 2024: 17191-17200.
> >
> > [4] Yao M, Richter O, Zhao G, et al. Spike-based dynamic computing with asynchronous sensing-computing neuromorphic chip[J]. Nature Communications, 2024, 15(1): 4464.

---

> > > ### Author Response · Authors · 2024-12-01
> > > **Official Comment by Authors**
> > >
> > > Dear Reviewer TRoC,
> > >
> > > We sincerely thank you for spending time and effort to review our manuscript and offer constructive feedback. As the discussion phase concludes, we hope our detailed responses have effectively addressed your concerns. If you need any additional clarifications or have unresolved questions, please feel free to contact us. We are more than willing to continue addressing your questions and enhancing our work based on your valuable insights.
> > >
> > > Best,
> > >
> > > Authors

---

### Official Review · Reviewer_P8Ri · 2024-11-02

**Soundness:** 2
**Presentation:** 3
**Contribution:** 2
**Rating:** 6
**Confidence:** 3

**Summary:**

This paper proposes an SNN-based single-shot multi-box detector SpikSSD. In the SpikSSD, this paper proposes a backbone MDS-ResNet. The MDS-ResNet adjusts the distribution of the input membrane synaptic to avoid gradient vanishing and explosion. Besides, this paper also proposes a Spiking Bi-direction Fusion Module (SBFM) to fuse spiking features in a bi-directional way. The proposed method is tested on GEN1 and VOC 2007 datasets, where the SpikSSD achieves good performance with only around 10% firing rate.

**Strengths:**

1. The proposed MDS-ResNet adjust uses tdBN in the shortcut to adjust the distribution of input membrane synaptic to alleviate the problems brought by gradient vanishing and explosion.
2. This paper gives a theoretical analysis of gradient vanishing/explosion for MDS-ResNet.

**Weaknesses:**

1. This paper claims they propose a backbone MDS-ResNet based on spiking neural networks. Is the backbone specifically designed for object detection? If so, what part of MDS-ResNet is specifically designed for objection detection? If not, it should be evaluated on classification datasets such as ImageNet [1] to show the generalization of the proposed backbone [2-5].
2. In the area of object detection, COCO [6] is a general dataset, why not evaluate the proposed method on this dataset?
3. What's the motivation for the proposed bi-directional fusion? What challenges are handled by the bi-directional fusion? It would be better to introduce more about that.
4. The experimental table of the paper shows the proposed method has a low energy. How is the low energy consumption realized? On general computational devices or specifically designed devices? If the low energy depends on specifically designed devices, how is the low energy realized?

[1] J. Deng, et al., Imagenet: A large-scale hierarchical image database. CVPR 2009.
[2] Y. Li, et al., Differentiable spike: Rethinking gradient-descent for training spiking neural networks. NeurIPS 2021.
[3] W. Fang, et al., Deep residual learning in spiking neural networks. NeurIPS 2021.
[3] H. Zheng, et al., Going deeper with directly-trained larger spiking neural networks. AAAI 2021.
[4] X. Yao, et al., GLIF: A unified gated leaky integrate-and-fire neuron for spiking neural networks. NeurIPS 2022.
[5]  W. Fang, et al., Parallel spiking neurons with high efficiency and ability to learn long-term dependencies. NeurIPS 2023.
[6] T. Lin, et al., Microsoft COCO: common objects in context. ECCV 2014.

**Questions:**

Please see the weaknesses part.

---

> ### Author Response · Authors · 2024-11-23
> **About Strengths**
>
> Thanks for your thoughtful review. We are pleased that you have recognized our superior performance on GEN1 and VOC datasets for SNN-based object detection. We would like to address your concerns and questions in the following sections.
>
> We would like to clarify that our proposed MDS-ResNet uses MDS in the shortcut to adjust the distribution of membrane synaptic input for subsequent neurons. This reduces the probability of extreme inputs, leading to better firing patterns, which thereby enhances the model's feature extraction capability. Rather than stabilizing gradients, our analysis of gradient vanishing or explosion is conducted to demonstrate that the MDS does not affect the training and convergence.

---

> > ### Author Response · Authors · 2024-11-23
> > **About Questions (Part 1/2)**
> >
> > Q1: This paper claims they propose a backbone MDS-ResNet based on spiking neural networks. Is the backbone specifically designed for object detection? If so, what part of MDS-ResNet is specifically designed for objection detection? If not, it should be evaluated on classification datasets such as ImageNet to show the generalization of the proposed backbone.
> >
> > A: We appreciate the reviewer's insights into MDS-ResNet. MDS-ResNet is proposed to address the insufficient feature extraction capability of current spiking backbone networks in object detection that involve both classification and regression. We propose Membrane-based Deformed Shortcut (MDS), a novel mechanism optimizing identity mapping to stabilize membrane synaptic input distribution across layers. This reduces the probability of extreme inputs, leading to better firing patterns, which thereby enhances the model's feature extraction capability. The experimental results in Tab. R3 also demonstrate the performance improvements of MDS-ResNet compared to other spiking networks in object detection.
> >
> > | Backbone         | Param(M) |   mAP    | mAP@0.5  | Firing Rate(%) | Energy(mJ) |
> > | :--------------- | :------: | :------: | :------: | :------------: | :--------: |
> > | DenseNet121-24   |   7.3    |   29.2   |   52.5   |     20.19      |    2.17    |
> > | SEW-ResNet18     |   9.8    |   32.2   |   58.2   |     10.46      |    1.92    |
> > | EMS-ResNet18     |   9.7    |   32.9   |   59.2   |     17.51      |    1.29    |
> > | MS-ResNet18      |   9.8    |   34.1   |   60.4   |     12.80      |    2.04    |
> > | **MDS-ResNet18** | **10.1** | **34.7** | **60.9** |   **12.41**    |  **0.97**  |
> >
> > **Table R3: Performance comparison between different spiking backbones on the GEN1 dataset.**
> >
> > However, we also recognize that MDS-ResNet enhances the feature extraction capability of spiking backbones, suggesting potential benefits for other applications as well. While this is beyond the scope of this study, we will explore these possibilities in our future work. We appreciate your interest and welcome you to follow our upcoming research.
> >
> > Q2: In the area of object detection, COCO is a general dataset, why not evaluate the proposed method on this dataset?
> >
> > A: Thanks for the suggestion. We have compared our method with other state-of-the-art approaches on the COCO 2017 dataset. Please refer to the section of To All Reviewers.
> >
> > Q3: What's the motivation for the proposed bi-directional fusion? What challenges are handled by the bi-directional fusion? It would be better to introduce more about that.
> >
> > A: We appreciate your thoughts on the proposed bi-directional fusion.
> > In SNNs, features exist in both spatial and temporal domains. Spiking feature fusion enhances feature representation in both dimensions, improving multi-scale detection capabilities. This dual-domain nature makes feature fusion more crucial in SNNs compared to ANNs.
> > Currently, the only fusion method designed for SNNs is the Spiking Fusion Module (SFM) [1]. This fusion method adopts a one-way, down-up approach, where large-scale features are upsampled to merge with small-scale features. Without up-down fusion, it only enhances small-scale feature representation, overlooking the potential improvement of large-scale features. Furthermore, SFM incorporates SEW-Block internally, which compromises the spiking nature of the network.
> > To address these issues, we propose the Spiking Bi-direction Fusion Module (SBFM). Specifically, this module utilizes Spiking Up/Down Block and membrane addition-based fusion method to process feature maps, performing fusion first bottom-up and then top-down. We employ the proposed MDS Fusion Block (MDSF-Block) to extract fused features during this process. These SNN-friendly components ensure SBFM adheres to spiking characteristics while enhancing the model detection capabilities.
> >
> > We have incorporated this analysis into the paper.

---

> > > ### Author Response · Authors · 2024-11-23
> > > **About Questions (Part 2/2)**
> > >
> > > Q4: The experimental table of the paper shows the proposed method has a low energy. How is the low energy consumption realized? On general computational devices or specifically designed devices? If the low energy depends on specifically designed devices, how is the low energy realized?
> > >
> > > A: We appreciate the reviewer's thoughts about energy consumption.
> > > In this paper, the energy consumption is calculated theoretically following common practices in SNN research [2, 3]. This method assumes 32-bit floating-point operations on general computational devices with 45nm technology [4], where $E_{MAC} = 4.6pJ\$ and $E_{AC} = 0.9pJ$. Detailed evaluation methods are provided in Appendix A.3.2.
> > >
> > > It’s worth noting that the above theoretical analysis excludes hardware architecture or data caching. The efficiency gains may be further amplified when SNNs are deployed on neuromorphic chips.
> > > These neuromorphic platforms, such as the Speck chip [5], operate in asynchronous mode without a global clock, leveraging their spike-driven characteristics. This design leads to minimal static energy consumption during periods without spike inputs. Consequently, the overall energy consumption of neuromorphic computing primarily scales with the number of spike-induced ACs [6].
> > >
> > > Although our focus is algorithmic innovation rather than hardware implementation, we acknowledge the need to discuss hardware implementation to demonstrate our energy advantages. This discussion has been added to the Discussion section of the Appendix.
> > >
> > > [1] Fan Y, Zhang W, Liu C, et al. SFOD: Spiking Fusion Object Detector[C]//Proceedings of the IEEE/CVF Conference on Computer Vision and Pattern Recognition. 2024: 17191-17200.
> > >
> > > [2] Hu Y, Deng L, Wu Y, et al. Advancing spiking neural networks toward deep residual learning[J]. IEEE Transactions on Neural Networks and Learning Systems, 2024.
> > >
> > > [3] Su Q, Chou Y, Hu Y, et al. Deep directly-trained spiking neural networks for object detection[C]//Proceedings of the IEEE/CVF International Conference on Computer Vision. 2023: 6555-6565.
> > >
> > > [4] Horowitz M. 1.1 computing's energy problem (and what we can do about it)[C]//2014 IEEE international solid-state circuits conference digest of technical papers (ISSCC). IEEE, 2014: 10-14.
> > >
> > > [5] Yao M, Richter O, Zhao G, et al. Spike-based dynamic computing with asynchronous sensing-computing neuromorphic chip[J]. Nature Communications, 2024, 15(1): 4464.
> > >
> > > [6] Pei J, Deng L, Song S, et al. Towards artificial general intelligence with hybrid Tianjic chip architecture[J]. Nature, 2019, 572(7767): 106-111.

---

> > > > ### Author Response · Authors · 2024-12-01
> > > > **Official Comment by Authors**
> > > >
> > > > Dear Reviewer P8Ri,
> > > >
> > > > We sincerely thank you for spending time and effort to review our manuscript and offer constructive feedback. As the discussion phase concludes, we hope our detailed responses have effectively addressed your concerns. If you need any additional clarifications or have unresolved questions, please feel free to contact us. We are more than willing to continue addressing your questions and enhancing our work based on your valuable insights.
> > > >
> > > > Best,
> > > >
> > > > Authors

---

> > > > > ### Comment · Reviewer_P8Ri · 2024-12-03
> > > > >
> > > > > Dear authors,
> > > > > Thank you for answering my questions. Most of my concerns have been addressed. I raise my score by 1.

---

### Official Review · Reviewer_ff5Q · 2024-11-03

**Soundness:** 2
**Presentation:** 3
**Contribution:** 2
**Rating:** 6
**Confidence:** 4

**Summary:**

This paper presents an MDS architecture for solving the problem of SNN target detection. This approach enables the model to further utilize spike operations to reduce computing power. The authors validated it on multiple target detection datasets.

**Strengths:**

1. The author's analysis of the spike operation in the algorithm is very accurate, and based on this, he designed a spike-friendly MDS architecture.
2. The method proposed by the authors has good performance and energy efficient. (more elegant than EMS method~)
3. The author provides a relatively complete theoretical analysis.

**Weaknesses:**

1. This article appears to be incremental work.
2. See Question part.

**Questions:**

1. Can you make a more detailed comparison of the EMS and MDS architectures in terms of formulas? And show why the EMS architecture cannot be fused to the spike operation?
2. Regarding the calculation of energy consumption, can you give a discussion on how to deploy it on real hardware? Similarly, I would like to see a comparison between MDS and other model architectures?
3. What do you think of the method of gradient identity analysis? Are there any problems with this method's series of assumptions in discrete models such as SNN?

---

> ### Author Response · Authors · 2024-11-23
> **About Contribution**
>
> We appreciate the reviewer's careful consideration of our work. We would like to formally clarify that our work presents two novel contributions specifically designed for SNNs.
>
> Firstly, we point out that the fundamental distinction between SNNs and ANNs is that features in SNNs are coded through neuron firing patterns. In traditional membrane-based shortcuts [1], the output of the residual path continuously accumulates to the shortcut path. Assuming these outputs are approximately independent, the variance of their sum equals the sum of their variances, leading to a gradual increase in membrane synaptic input variance for subsequent neurons. This results in a higher probability of extreme inputs. Specifically, when the synaptic input is small, the accumulated membrane potential fails to reach the firing threshold. When it is excessively large, neurons exhibit abnormally high firing rates. In both cases, the firing pattern becomes either overly sparse or excessive, failing to reflect the input features accurately. This unstable firing pattern makes it difficult for the network to distinguish and extract meaningful features, which in turn affects the network’s feature extraction capability.
>
> To address this issue, we propose the Membrane-based Deformed Shortcut (MDS) mechanism, which innovatively stabilizes membrane synaptic input distribution across layers. Based on MDS, we propose a novel Spiking Residual Network, MDS-ResNet, which exhibits stronger feature extraction capabilities. Extensive experimental results, as shown in Tab. R3, demonstrate the superiority of MDS-ResNet over other spiking backbones in object detection tasks.
> This finding reveals the crucial role of membrane synaptic input distribution in SNN feature extraction, providing new insights and directions for future SNN development.
>
> | Backbone         | Param(M) |   mAP    | mAP@0.5  | Firing Rate(%) | Energy(mJ) |
> | :--------------- | :------: | :------: | :------: | :------------: | :--------: |
> | DenseNet121-24   |   7.3    |   29.2   |   52.5   |     20.19      |    2.17    |
> | SEW-ResNet18     |   9.8    |   32.2   |   58.2   |     10.46      |    1.92    |
> | EMS-ResNet18     |   9.7    |   32.9   |   59.2   |     17.51      |    1.29    |
> | MS-ResNet18      |   9.8    |   34.1   |   60.4   |     12.80      |    2.04    |
> | **MDS-ResNet18** | **10.1** | **34.7** | **60.9** |   **12.41**    |  **0.97**  |
>
> **Table R3: Performance comparison between different spiking backbones on the GEN1 dataset.**
>
> Secondly, the Spiking Bi-direction Fusion Module (SBFM) represents the first spiking bi-directional feature fusion implementation in SNNs.
> Currently, research in feature fusion for SNNs is limited to one-way down-up fusion [2], where large-scale features are only upsampled to enhance small-scale features. This approach restricts up-down information transmission and leads to inadequate feature fusion across scales. Furthermore, this method fails to preserve inherent spiking characteristics.
> To address these issues, we propose the Spiking Bi-direction Fusion Module (SBFM). Specifically, this module utilizes Spiking Up/Down Block and membrane addition-based fusion method to process feature maps, performing fusion first bottom-up and then top-down. We employ the proposed MDS Fusion Block (MDSF-Block) to extract fused features during this process. These SNN-friendly components ensure SBFM adheres to spiking characteristics while enhancing the model detection capabilities.
>
> In summary, our research provides new insights for the entire field of SNNs, rather than being incremental work. Specifically, we highlight two breakthroughs: (1) membrane synaptic input distribution is crucial for feature extraction capability in SNNs, and (2) spiking bi-directional feature fusion in SNNs can achieve stronger feature integration compared to one-way fusion approaches. Moreover, we have provided experimental validation for both innovations, demonstrating their effectiveness in enhancing SNN performance.

---

> > ### Author Response · Authors · 2024-11-23
> > **About Questions (Part 1/2)**
> >
> > Thank you for your thoughtful review. We are pleased that you have recognized our analysis of the spike operation in the algorithm is very accurate. We would like to address your concerns and questions in the following sections.
> >
> > Q1: Can you make a more detailed comparison of the EMS and MDS architectures in terms of formulas? And show why the EMS architecture cannot be fused to the spike operation?
> >
> > A: We appreciate the reviewer's thoughts on our work. For EMS-ResNet, its residual learning can be expressed as follows.
> > \begin{equation}
> > \psi = \mathrm{tdBN} \circ \mathrm{3x3Conv} \circ \mathrm{SN}
> > \end{equation}
> > \begin{equation}
> > Y_{I}^{L}=\psi\left(\psi(X^L)\right) + X^L
> > \end{equation}
> > \begin{equation}
> > Y_{D1}^{L}=\psi\left(\psi (X^L)\right)+\mathrm{Concat}[\mathrm{MaxPool}(X^L),\psi(\mathrm{MaxPool}(X^L))]
> > \end{equation}
> > \begin{equation}
> > Y_{D2}^{L}=\psi\left(\psi(X^L)\right)+\psi(\mathrm{MaxPool}(X^L))
> > \end{equation}
> >
> > Here, $ Y $ and $ X $ denote output and input, respectively. The superscript $ L $ represents the $ L $-th residual block. The subscripts indicate different dimensional changes: $ D1 $ for downsampling with increased channels, $ D2 $ for downsampling with unchanged or decreased channels, and $ I $ for no downsampling. $ \mathrm{SN} $, $ \mathrm{tdBN} $, $ \mathrm{3x3Conv} $, and $ \mathrm{MaxPool} $ represent spiking neurons, threshold-dependent Batch Normalization, convolution, and Maxpool operations.
> >
> > However, we believe that the $ Y_I^L $ in EMS-ResNet does not fully consider the SNN characteristics.
> > For ease of analysis, let $ Y_R^L $ denote the residual path output and $ Y_S^L $ denote the shortcut path output. We assume the distributions of these two variables to be $ Y_R^L \sim \mathcal{N}(0, {\sigma_R^L}^2) $ and $ Y_S^L \sim \mathcal{N}(0, {\sigma_S^L}^2) $, where $ {\sigma_R^L}^2 $ and $ {\sigma_S^L}^2 $ represent their respective variances. Assuming the two variables are approximately independent, we can derive $ Y_I^L \sim \mathcal{N}(0, {\sigma_R^L}^2+{\sigma_S^L}^2) $. However, there is no computation in both the shortcut path of $ Y_I^L $ and between individual residual blocks in EMS-ResNet. Therefore, ${\sigma_S^L}^2$ can be further decomposed into ${\sigma_R^{L-1}}^2+{\sigma_S^{L-1}}^2$, and consequently, we can derive that $ Y_I^L \sim \mathcal{N}(0, {\sigma_R^L}^2+{\sigma_R^{L-1}}^2+{\sigma_S^{L-1}}^2) $.
> > As the number of stacked $Y_{I}^{L}$ blocks increases, the membrane synaptic input distribution in subsequent neurons becomes increasingly unstable. Consequently, the probability of overly sparse or excessive firing patterns increases, which impedes accurate feature representation and ultimately weakens the model's feature extraction capability.
> >
> > To address this issue, we propose the Membrane-based Deformed Shortcut (MDS) to enhance $Y_{I}^{L}$. The improved $Y_{I^{*}}^{L}$ can be expressed as follows.
> >
> > \begin{equation}
> > \phi = \mathrm{tdBN} \circ \mathrm{1x1Conv} \circ \mathrm{SN}
> > \end{equation}
> >
> > \begin{equation}
> > Y_{I^{*}}^{L}=\psi\left(\psi(X^L)\right) + \phi(X^L)
> > \end{equation}
> >
> > Where $\phi$ represents MDS, $\mathrm{SN}$ layer converts input into spikes, followed by a $\mathrm{1x1Conv}$ that enhances the network's expressiveness, and the introduction of $\mathrm{tdBN}$ enables the shortcut path to adjust its output distribution. Based on $Y_{I^{*}}^{L}$, we propose MDS-ResNet. Our analysis is supported by both quantitative results in Tab. R4, showing superior performance of MDS-ResNet, and qualitative results in Fig. 1 of our paper, demonstrating improved firing patterns.
> >
> > | Backbone         | Param(M) |   mAP    | mAP@0.5  | Firing Rate(%) | Energy(mJ) |
> > | :--------------- | :------: | :------: | :------: | :------------: | :--------: |
> > | EDS-ResNet18     |   9.7    |   32.9   |   59.2   |     17.51      |    1.29    |
> > | **MDS-ResNet18** | **10.1** | **34.7** | **60.9** |   **12.41**    |  **0.97**  |
> >
> > **Table R4: Performance comparison between MDS-ResNet and EMS-ResNet on the GEN1 dataset.**
> >
> > Furthermore, we want to clarify that we never claim EMS architecture cannot be fused with spike operations. Instead, the key limitation of EMS-ResNet lies in its insufficient consideration of SNN characteristics, resulting in increasingly unstable membrane synaptic input distribution in deeper networks.

---

> > > ### Author Response · Authors · 2024-11-23
> > > **About Questions (Part 2/2)**
> > >
> > > Q2: Regarding the calculation of energy consumption, can you give a discussion on how to deploy it on real hardware? Similarly, I would like to see a comparison between MDS and other model architectures?
> > >
> > > A: We appreciate the reviewer's thoughts about the energy consumption of hardware deployment. We would like to clarify that our work primarily focuses on algorithmic innovations rather than hardware implementation. The discussion regarding hardware implementation has been added to the Discussion section of the Appendix.
> > >
> > > In this paper, we have conducted theoretical energy evaluations to compare energy consumption between ANNs and SNNs, following common practices in SNN research [2, 3]. Our method shows substantial energy advantages compared to both SNN and ANN solutions. Detailed evaluation methods are provided in Appendix A.3.2.
> > >
> > > It’s worth noting that the above theoretical analysis excludes hardware architecture or data caching. The efficiency gains may be further amplified when SNNs are deployed on neuromorphic chips.
> > > These neuromorphic platforms, such as the Speck chip [4], operate in asynchronous mode without a global clock, leveraging their spike-driven characteristics. This design leads to minimal static energy consumption during periods without spike inputs. Consequently, the overall energy consumption of neuromorphic computing primarily scales with the number of spike-induced ACs [5].
> > >
> > > For deployment to real hardware, we take the neuromorphic chip Speck as an example. First, we need to use Sinabs, a PyTorch-based deep learning library, to build and train SNN models. Through the Sinabs-Speck plugin, we can complete model quantization and configuration generation. Subsequently, we can use the Samna tool for deployment and execution.
> > >
> > > Compared to other SNN-based methods, our SpikSSD has lower firing rates, thus leading to reduced static energy consumption when deployed on neuromorphic chips. Therefore, it will demonstrate greater advantages than theoretical energy evaluations.
> > >
> > > Q3: What do you think of the method of gradient identity analysis? Are there any problems with this method's series of assumptions in discrete models such as SNN?
> > >
> > > A: In this paper, we utilize Block Dynamical Isometry [6] to conduct gradient identity analysis. This method establishes a modular statistical framework based on free probability theory, enabling its broad application to various network architectures with easily verifiable prerequisites.
> > >
> > > The application of Block Dynamical Isometry for gradient identity analysis in SNNs is appropriate. The key characteristic of SNNs lies in their use of spiking neurons, such as LIF, for computation.
> > > As proven by [7], the LIF layer satisfies the general linear transformation requirement in Block Dynamical Isometry. In this paper, we primarily use LIF and PLIF neurons (which only makes the membrane time constant in LIF learnable without changing LIF's computation mechanism, thus the same proof applies). Therefore, this method is suitable for analyzing our proposed SNN model and has been widely adopted in numerous SNN models [1, 3, 7].
> > >
> > > [1] Hu Y, Deng L, Wu Y, et al. Advancing spiking neural networks toward deep residual learning[J]. IEEE Transactions on Neural Networks and Learning Systems, 2024.
> > >
> > > [2] Fan Y, Zhang W, Liu C, et al. SFOD: Spiking Fusion Object Detector[C]//Proceedings of the IEEE/CVF Conference on Computer Vision and Pattern Recognition. 2024: 17191-17200.
> > >
> > > [3] Su Q, Chou Y, Hu Y, et al. Deep directly-trained spiking neural networks for object detection[C]//Proceedings of the IEEE/CVF International Conference on Computer Vision. 2023: 6555-6565.
> > >
> > > [4] Yao M, Richter O, Zhao G, et al. Spike-based dynamic computing with asynchronous sensing-computing neuromorphic chip[J]. Nature Communications, 2024, 15(1): 4464.
> > >
> > > [5] Pei J, Deng L, Song S, et al. Towards artificial general intelligence with hybrid Tianjic chip architecture[J]. Nature, 2019, 572(7767): 106-111.
> > >
> > > [6] Chen Z, Deng L, Wang B, et al. A comprehensive and modularized statistical framework for gradient norm equality in deep neural networks[J]. IEEE Transactions on Pattern Analysis and Machine Intelligence, 2020, 44(1): 13-31.
> > >
> > > [7] Zheng H, Wu Y, Deng L, et al. Going deeper with directly-trained larger spiking neural networks[C]//Proceedings of the AAAI conference on artificial intelligence. 2021, 35(12): 11062-11070.

---

> > > ### Comment · Reviewer_ff5Q · 2024-11-26
> > >
> > > I think the author clearly stated their contributions and all my questions are well answered. I think it is worth to raise my score by 1.

---

### Official Review · Reviewer_wnbY · 2024-11-03

**Soundness:** 2
**Presentation:** 2
**Contribution:** 2
**Rating:** 3
**Confidence:** 4

**Summary:**

The paper presents SpikSSD, an SNN-based object detection network. SpikSSD introduces a spiking residual network (MDS-ResNet) for feature extraction and Spiking Bi-direction Fusion Module (SBFM) for multi-scale feature fusion. Experiments on GEN1 and VOC dataset indicates the best performance of SNN-based object detection.

**Strengths:**

1.SpikSSD shows improvements in energy efficiency, achieving a 10% firing rate with competitive accuracy, which is beneficial for power-constrained applications.
2.Sufficient mathematical proof of the network structure mathematically, which proves the liability of the proposed MDS-ResNet.
3.Experiments on GEN1 and VOC proves the performance of SpikeSSD. On both dataset SpikeSSD proves to be the best SNN-based object detection network.

**Weaknesses:**

1.For object detection on images, the authors only provide experiment results on VOC dataset, which is insufficient since MS-COCO dataset is the mainstream dataset for this task. VOC dataset has much fewer images, which lacks of utility. There should be experiments on MS-COCO dataset and real-world images for validation qualitatively.

2.SBFM is too complicated for SNN-based network. SBFM fuses different scales of feature maps, but it is too complicated for training. To meet the requirement of SSD detection head, multiple features need to be generated. SBFM has a PAN-like structure but with 5 feature maps. There could be a more efficient way of feature fusion other than the proposed SBFM.

3.The MDS-ResNet primarily extends SNN modules on mature ANN networks, which lacks new design or new insight on SNN. Core ideas like membrane-based shortcuts and spiking feature fusion have seen prior application.

SpikSSD provides a model that is interesting but the contributions are incremental, offering refinements on existing modules rather than making novel breakthroughs. Additionally, without real-world experiments, it is hard to identify the claimed efficiency gains. The paper would benefit from hardware validation and further simplification for real-world deployment or application, rather than merely experimenting on certain dataset.

**Questions:**

1.There could be experiments on MS-COCO dataset and real-world images for validation qualitatively.

2. Could explain the following part with efficient manner? "SBFM has a PAN-like structure but with 5 feature maps. There could be a more efficient way of feature fusion other than the proposed SBFM."

---

> ### Author Response · Authors · 2024-11-23
> **About Contribution**
>
> We appreciate the reviewer's careful consideration of our work. We would like to formally clarify that our work presents two novel contributions specifically designed for SNNs.
>
> Firstly, we point out that the fundamental distinction between SNNs and ANNs is that features in SNNs are coded through neuron firing patterns. In traditional membrane-based shortcuts [1], the output of the residual path continuously accumulates to the shortcut path. Assuming these outputs are approximately independent, the variance of their sum equals the sum of their variances, leading to a gradual increase in membrane synaptic input variance for subsequent neurons. This results in a higher probability of extreme inputs. Specifically, when the synaptic input is small, the accumulated membrane potential fails to reach the firing threshold. When it is excessively large, neurons exhibit abnormally high firing rates. In both cases, the firing pattern becomes either overly sparse or excessive, failing to reflect the input features accurately. This unstable firing pattern makes it difficult for the network to distinguish and extract meaningful features, which in turn affects the network’s feature extraction capability.
>
> To address this issue, we propose the Membrane-based Deformed Shortcut (MDS), a novel mechanism optimizing identity mapping to stabilize membrane synaptic input distribution across layers. Based on MDS, we propose a novel Spiking Residual Network, MDS-ResNet, which exhibits stronger feature extraction capabilities. Extensive experimental results, as shown in Tab. 1 of our paper, demonstrate the superiority of MDS-ResNet over other spiking backbones in object detection tasks. This solution is specifically designed for SNN characteristics rather than simply extending SNN modules on mature ANN networks, providing new insights for future SNN development.
>
> Secondly, the Spiking Bi-direction Fusion Module (SBFM) represents the first spiking bi-directional feature fusion implementation in SNNs.
> Currently, research in feature fusion for SNNs is limited to one-way down-up fusion [2], where large-scale features are only upsampled to enhance small-scale features. This approach restricts up-down information transmission and leads to inadequate feature fusion across scales. Furthermore, this method fails to preserve inherent spiking characteristics.
> To address these issues, we propose the SBFM which achieves comprehensive bi-directional feature integration while maintaining pure spike-based computation.
> Notably, we have developed SBFM based on intrinsic characteristics of SNNs, rather than following fusion methods from mature ANN models like YOLO [3] and PAN [4]. Our design incorporates SNN-specific innovations, including the membrane addition-based fusion method, MDSF-Block, and Spiking up/down block, each specifically designed to leverage the unique properties of SNNs.
>
> These contributions stem from analyzing intrinsic SNN characteristics rather than extending mature ANN architectures. Our work provides new insights for SNN design (e.g., highlighting that membrane synaptic input distribution is crucial for feature extraction capability in SNNs) instead of merely making refinements to existing modules. Moreover, we have provided experimental validation for both innovations, demonstrating their effectiveness in enhancing SNN performance.

---

> > ### Author Response · Authors · 2024-11-23
> > **About Questions**
> >
> > Thanks for your thoughtful review. We are pleased that you have recognized our superior performance on GEN1 and VOC datasets for SNN-based object detection. We would like to address your concerns and questions in the following sections.
> >
> > Q1: There could be experiments on MS-COCO dataset and real-world images for validation qualitatively.
> >
> > A: Thanks for the suggestion. We have compared our method with other state-of-the-art approaches on the COCO 2017 dataset. Please refer to the section of To All Reviewers.
> >
> > Q2: Could explain the following part with efficient manner? "SBFM has a PAN-like structure but with 5 feature maps. There could be a more efficient way of feature fusion other than the proposed SBFM."
> >
> > A: We sincerely appreciate the reviewer's thoughtful consideration of SBFM. First, we would like to clarify that although both SBFM and PAN achieve bi-directional fusion, they are fundamentally different in their implementation, design principles, and objectives.
> > Unlike PAN which uses 3×3 convolutions and addition operations to enhance feature pyramids, SBFM introduces SNN-specific innovations for spiking bi-directional feature fusion, an unexplored area in the field. These innovations, including membrane addition-based fusion method, MDSF-Block, and SNN-friendly up/down block, achieve comprehensive feature integration while maintaining pure spike-based computation.
> >
> > Regarding efficiency, we believe SBFM's complexity is appropriate and necessary for SNNs. Given that SNNs inherently have limited feature representation capability due to their spike-based computation nature, incorporating multiple-scale features helps enhance multi-scale detection performance. To validate this, we conduct additional experiments (shown in rows 1-3 of Tab. R2) with simplified versions of SBFM using 3 or 4 feature maps. The results demonstrate that such simplifications lead to significant performance degradation while offering minimal parameter reduction. Furthermore, as shown in Tab. R2 (rows 4-5), downgrading SBFM to one-way feature fusion or simplifying MDSF-Block also results in substantial performance drops. These findings support our design choices for SBFM's complexity.
> >
> > While we acknowledge that more efficient spike-based feature fusion methods may emerge as the field of SNNs evolves (similar to PAN's optimization in ANNs), our extensive experiments demonstrate that SBFM currently achieves the best spiking feature fusion performance for SNN-based object detection.
> >
> > | Backbone         |  Fusion  | Fusion Layers | Param(M) |   mAP    | mAP@0.5  | Firing Rate(%) | Energy(mJ) |
> > | :--------------- | :------: | :-----------: | :------: | :------: | :------: | :------------: | :--------: |
> > | MDS-ResNet18     |   SBFM   |       3       |   8.2    |   33.9   |   59.9   |     10.06      |    1.33    |
> > | MDS-ResNet18     |   SBFM   |       4       |   12.2   |   36.2   |   62.7   |     10.70      |    1.48    |
> > | **MDS-ResNet18** | **SBFM** |     **5**     | **13.9** | **37.6** | **64.3** |   **10.57**    |  **1.47**  |
> > | MDS-ResNet18     |   SSFM   |       5       |   12.5   |   36.5   |   63.2   |     11.01      |    1.43    |
> > | MDS-ResNet18     |  SBFM*   |       5       |   12.1   |   36.9   |   63.6   |     11.32      |    1.25    |
> >
> > **Table R2: Ablation studies on SBFM complexity using the GEN1 dataset.** A * indicates that Spiking Depthwise Separable Convolution is not used in SBFM.

---

> > > ### Author Response · Authors · 2024-11-23
> > > **About Weakness**
> > >
> > > Q1: For object detection on images, the authors only provide experiment results on VOC dataset, which is insufficient since MS-COCO dataset is the mainstream dataset for this task. VOC dataset has much fewer images, which lacks of utility. There should be experiments on MS-COCO dataset and real-world images for validation qualitatively.
> > >
> > > A: Thanks for the suggestion. We have compared our method with other state-of-the-art approaches on the COCO 2017 dataset. Please refer to the section of To All Reviewers.
> > >
> > > Q2: SBFM is too complicated for SNN-based network. SBFM fuses different scales of feature maps, but it is too complicated for training. To meet the requirement of SSD detection head, multiple features need to be generated. SBFM has a PAN-like structure but with 5 feature maps. There could be a more efficient way of feature fusion other than the proposed SBFM.
> > >
> > > A: We appreciate the reviewer's concern about the efficiency of SBFM.
> > > Regarding training complexity, our experimental results show that SBFM actually facilitates network convergence, contrary to concerns about increased training difficulty.
> > > As shown in Fig. 6 of the Appendix, with the same training hyperparameters, MDS-ResNet18 with SBFM achieves faster and better convergence compared to its counterpart without SBFM.
> > > We attribute this improvement to SBFM’s construction of two shorter backpropagation paths from the deepest features to the input, significantly benefiting training and convergence.
> > >
> > > For the remaining questions, please refer to Section Q2 in "About Questions" for a detailed discussion of SBFM's feature fusion design and efficiency.
> > >
> > > Q3: The MDS-ResNet primarily extends SNN modules on mature ANN networks, which lacks new design or new insight on SNN. Core ideas like membrane-based shortcuts and spiking feature fusion have seen prior application.
> > >
> > > A: We appreciate the reviewer's careful consideration of our work. We disagree with this assessment and would like to clarify our key contributions. Our innovations stem from analyzing intrinsic SNN properties rather than simply extending mature ANN networks. Specifically, the MDS mechanism addresses the membrane synaptic input instability issue in SNNs, while SBFM achieves full-spiking bi-directional feature fusion for the first time. For a more detailed discussion of these contributions, please refer to the "About Contributions" section.
> > >
> > > Q4: SpikSSD provides a model that is interesting but the contributions are incremental, offering refinements on existing modules rather than making novel breakthroughs.
> > >
> > > A: We appreciate the reviewer's consideration of our work. We disagree that our work merely offers refinements on existing modules. Instead, our research provides new insights for the entire SNNs field by highlighting two key findings: (1) membrane synaptic input distribution is crucial for feature extraction capability in SNNs, and (2) spiking bi-directional feature fusion in SNNs can achieve stronger feature integration compared to one-way fusion approaches. For a more detailed discussion of these contributions, please refer to the "About Contributions" section.
> > >
> > > Q5: Additionally, without real-world experiments, it is hard to identify the claimed efficiency gains. The paper would benefit from hardware validation and further simplification for real-world deployment or application, rather than merely experimenting on certain dataset.
> > >
> > > A: We appreciate the reviewer's suggestion about hardware validation. We would like to clarify that our work primarily focuses on algorithmic innovations rather than hardware implementation. In this paper, we have conducted theoretical energy evaluations to compare energy consumption between ANNs and SNNs, following common practices in SNN research [2, 5]. Our method shows substantial energy advantages compared to both SNN and ANN solutions. Detailed evaluation methods are provided in Appendix A.3.2. The discussion regarding hardware implementation has been added to the Discussion section in the Appendix of our paper.
> > >
> > > It’s worth noting that the above theoretical analysis excludes hardware architecture or data caching. The efficiency gains may be further amplified when SNNs are deployed on neuromorphic chips.
> > > These neuromorphic platforms, such as the Speck chip [6], operate in asynchronous mode without a global clock, leveraging their spike-driven characteristics. This design leads to minimal static energy consumption during periods without spike inputs. Consequently, the overall energy consumption of neuromorphic computing primarily scales with the number of spike-induced ACs [7].

---

> > > > ### Author Response · Authors · 2024-11-23
> > > > **References**
> > > >
> > > > [1] Hu Y, Deng L, Wu Y, et al. Advancing spiking neural networks toward deep residual learning[J]. IEEE Transactions on Neural Networks and Learning Systems, 2024.
> > > >
> > > > [2] Fan Y, Zhang W, Liu C, et al. SFOD: Spiking Fusion Object Detector[C]//Proceedings of the IEEE/CVF Conference on Computer Vision and Pattern Recognition. 2024: 17191-17200.
> > > >
> > > > [3] Wang A, Chen H, Liu L, et al. Yolov10: Real-time end-to-end object detection[J]. arXiv preprint arXiv:2405.14458, 2024.
> > > >
> > > > [4] Liu S, Qi L, Qin H, et al. Path aggregation network for instance segmentation[C]//Proceedings of the IEEE conference on computer vision and pattern recognition. 2018: 8759-8768.
> > > >
> > > > [5] Su Q, Chou Y, Hu Y, et al. Deep directly-trained spiking neural networks for object detection[C]//Proceedings of the IEEE/CVF International Conference on Computer Vision. 2023: 6555-6565.
> > > >
> > > > [6] Yao M, Richter O, Zhao G, et al. Spike-based dynamic computing with asynchronous sensing-computing neuromorphic chip[J]. Nature Communications, 2024, 15(1): 4464.
> > > >
> > > > [7] Pei J, Deng L, Song S, et al. Towards artificial general intelligence with hybrid Tianjic chip architecture[J]. Nature, 2019, 572(7767): 106-111.

---

> > > > > ### Author Response · Authors · 2024-12-01
> > > > > **Official Comment by Authors**
> > > > >
> > > > > Dear Reviewer wnbY,
> > > > >
> > > > > We sincerely thank you for spending time and effort to review our manuscript and offer constructive feedback. As the discussion phase concludes, we hope our detailed responses have effectively addressed your concerns. If you need any additional clarifications or have unresolved questions, please feel free to contact us. We are more than willing to continue addressing your questions and enhancing our work based on your valuable insights.
> > > > >
> > > > > Best,
> > > > >
> > > > > Authors

---

### Author Response · Authors · 2024-11-23
**To All Reviewers**

We sincerely thank all reviewers for their valuable feedback and suggestions on our paper. Here, we would like to address the concerns regarding the performance of SpikSSD on the COCO 2017 dataset.

On the COCO 2017 dataset, as demonstrated in Tab. R1, SpikSSD not only achieves the second-highest mAP@0.5 among all SNN-based methods but also ranks first among directly-trained SNN-based methods while consuming less energy.
Notably, while SUHD achieves slightly better results on COCO 2017, our approach outperforms it on the VOC 2007 dataset, demonstrating the overall superiority of SpikSSD across different frame-based datasets.
Moreover, when compared to ANN-based methods, SpikSSD-L demonstrates comparable performance while requiring merely 1/3 (YOLOV5s) or even 1/37 (DETR) of the energy consumption.

| Method                | Model           | Param(M) | mAP@0.5  | Energy(mJ) |
| :-------------------- | :-------------- | :------: | :------: | :--------: |
| ANNs                  | YOLOv5s         |   7.3    |   54.8   |   38.41†   |
| ANNs                  | **DETR**        | **41.0** | **55.7** | **430.10** |
| ANNs to SNNs          | Vthfast+Vthacc  |   8.9    |   21.1   |     -      |
| ANNs to SNNs          | Spiking-YOLO    |   8.9    |   25.7   |     -      |
| ANNs to SNNs          | SpiCalib        |   23.9   |   45.4   |     -      |
| ANNs to SNNs          | **SUHD**        |  **-**   | **54.6** |   **-**    |
| Directly trained SNNs | Hybrid-YOLO     |    -     |   26.2   |     -      |
| Directly trained SNNs | EMS-YOLO        |   26.9   |   50.1   |   29.00    |
| Directly trained SNNs | **SpikeFormer** | **75.0** | **51.2** | **140.80** |
| Directly trained SNNs | **SpikSSD-S**   | **16.3** | **49.2** | **11.65**  |
| Directly trained SNNs | **SpikSSD-L**   | **21.3** | **52.4** | **15.59**  |

**Table R1: Comparison with state-of-the-art models on the COCO 2017 dataset.** † indicates that energy is recalculated using our energy consumption method for fair comparison.

---

### Meta-Review · Area_Chair_Rzd6 · 2024-12-20

**Metareview:**

This paper proposes SpikSSD, a Spiking Single Shot Multibox Detector. It leverages a full-spiking backbone network, MDS-ResNet, and a Spiking Bi-direction Fusion Module (SBFM). Experimental results show that SpikSSD outperforms existing SNN-based approaches with a low firing rate.

After the rebuttal period, 5 reviewers rate 3, 3, 3, 6, 6, respectively. All reviewers agree that the proposed SpikSSD has excellent performance and low energy consumption, which are the strengths of this paper. However, there are some criticisms. Reviewer wnbY thinks that the contributions are incremental. Reviewer TRoC thinks that the mechanisms of MDS and SBFM lack clarity. Reviewer V2pm believes that the motivation of MDS-ResNet is too complex. I think these criticisms are justified. In my opinion, this paper is not well-written. I suggest that the authors further clarify the proposed mechanisms and highlight the contributions. For example, it would be helpful to add a figure comparing with the existing methods which do not have bidirectional fusion to highlight the contribution of SBFM.

While the authors argue that Reviewers TRoC and V2pm overlooked the key contributions of the proposed SBFM and focused on MDS-ResNet, I believe this may be due to poor content organization. As Reviewer V2pm states, "The authors spend a lot of words introducing the proposed backbone MDS-ResNet". Sections 3.2 and 3.3 are significantly longer than Section 3.4, making the paper focus on MDS-ResNet rather than SBFM. I suggest that the authors reorganize the paper to highlight the contribution of SBFM.

Overall, I believe this paper would benefit from further revision. Therefore, the final decision is to reject this paper.

**Additional Comments On Reviewer Discussion:**

After the rebuttal period, Reviewers ff5Q and P8Ri find their concerns addressed and raise their ratings to 6, while other reviewers keep the original rating of 3. Reviewer V2pm thinks the concerns remain. I agree that there are still some problems with this paper after the rebuttal period.

---

### Decision · Program_Chairs · 2025-01-22

Reject